# Mutant mice with rod-specific VPS35 deletion exhibit retinal α-synuclein pathology-associated degeneration

Cheng Fu[1,10], Nan Yang [1,10], Jen-Zen Chuang [1,10], Nobuyuki Nakajima [1,7,10], Satoshi Iraha[1,8,10], Neeta Roy[1], Zhenquan Wu[1], Zhichun Jiang[2], Wataru Otsu [1,9], Roxana A. Radu [2], Howard Hua Yang [3], Maxwell Ping Lee [3], Tilla S. Worgall [4], Wen-Cheng Xiong[5] & Ching-Hwa Sung [1,6] ✉

Vacuolar protein sorting 35 (VPS35), the core component of the retromer complex which regulates endosomal trafficking, is genetically linked with Parkinson's disease (PD). Impaired vision is a common non-motor manifestation of PD. Here, we show mouse retinas with VPS35-deficient rods exhibit synapse loss and visual deficit, followed by progressive degeneration concomitant with the emergence of Lewy body-like inclusions and phospho-α-synuclein (P-αSyn) aggregation. Ultrastructural analyses reveal VPS35-deficient rods accumulate aggregates in late endosomes, deposited as lipofuscins bound to P-αSyn. Mechanistically, we uncover a protein network of VPS35 and its interaction with HSC70. VPS35 deficiency promotes sequestration of HSC70 and P-αSyn aggregation in late endosomes. Microglia which engulf lipofuscins and P-αSyn aggregates are activated, displaying autofluorescence, observed as bright dots in fundus imaging of live animals, coinciding with pathology onset and progression. The Rod$^{\Delta Vps35}$ mouse line is a valuable tool for further mechanistic investigation of αSyn lesions and retinal degenerative diseases.

Parkinson's disease (PD) is the second most common neurodegeneration disease. However, non-invasive early diagnostic biomarkers and effective treatment of PD remain unmet needs. More than 80% of PD patients develop early visual symptoms (e.g., impaired visual acuity, spatial contrast sensitivity, depth perception, and color vision)[1–3]. Also, retinal thinning due to the loss of retinal neurons is seen in early human PD[4–9] and several rodent models of PD[3,7,10].

Though the mechanism underlying retinal cell death in PD is unknown, retinal thickness, determined by spectrum-domain optical coherence tomography (SD-OCT), has been used as a biomarker during human trials of PD treatments[7]. Lewy body (LB) inclusions are the pathologic hallmark of PD and have been observed in PD patient retinas[11,12]. α-synuclein (αSyn), particularly the phospho(S219)-αSyn (herein P-αSyn[13–15]), is highly expressed in LB inclusions. A positive

[1]Department of Ophthalmology, Margaret M. Dyson Vision Research Institute, Weill Cornell Medicine, 1300 York Avenue, New York, NY 10065, USA. [2]UCLA Stein Eye Institute, and Department of Ophthalmology, David Geffen School of Medicine at UCLA, Los Angeles, CA, USA. [3]The Laboratory of Cancer Biology and Genetics, Center for Cancer Research, National Cancer Institute, National Institutes of Health, Bethesda, MD, USA. [4]Department of Pathology and Cell Biology, Columbia University Medical Center, New York, NY, USA. [5]Department of Neurosciences, School of Medicine, Case Western Reserve University, Cleveland, OH, USA. [6]Department of Cell and Developmental Biology, Weill Cornell Medicine, 1300 York Avenue, New York, NY 10065, USA. [7]Present address: Department of Urology, Tokai University School of Medicipne, Tokyo, Japan. [8]Present address: Department of Ophthalmology, Faculty of Life Sciences, Kumamoto University; Department of Ophthalmology, National Sanatorium Kikuchi Keifuen, Kumamoto, Japan. [9]Present address: Department of Biomedical Research Laboratory, Gifu Pharmaceutical University, Gifu, Japan. [10]These authors contributed equally: Cheng Fu, Nan Yang, Jen-Zen Chuang, Nobuyuki Nakajima, Satoshi Iraha. ✉e-mail: chsung@med.cornell.edu

correlation between the density of P-αSyn-labeled inclusions/neurites in retinal ganglion cells of postmortem PD patient eyes and their clinical grading scale has been reported[12]. αSyn, which is important for synaptic vesicle (SV) recycling, is especially susceptible to misfolding and forming amorphous aggregates and fibrils[16–20]. Also critical for SV recycling is the molecular chaperon HSC70 (or HSPA8). HSC70 uncoats clathrin-coated vesicles in presynaptic terminals[21], facilitates the proper folding of newly translated proteins[22], and escorts misfolded proteins, such as αSyn, for selective autophagic degradation[22–29].

The mammalian retina has an organized laminated structure; the soma and synapses of 6 major types of neurons are arranged into distinct layers (Fig. 1a). The rod photoreceptors have four compartments: outer segment (OS), inner segment (IS), cell body, and synaptic terminal. These are organized into the OS, IS, outer nuclear (ONL), and outer plexiform (OPL) layers, respectively, in the outer part of the retina. The OS and synaptic components are synthesized in the IS and then transported to their final destinations. It has been well established that the degradation of OS components relies on the neighboring retinal pigment epithelial (RPE) cells through daily phagocytosis of the shed distal OS tips[30]. In contrast, little is known about the cell-autonomous mechanisms by which photoreceptors remove the molecules that are mislocalized, either due to protein misfolding or mistrafficking. Improper protein folding and trafficking are common defects shared by several mouse models of retinitis pigmentosa

(RP)[30,31]. RP is the most common form of inherited retinal degenerative disease characterized by primary rod cell death.

Emerging studies show that the endolysosomal system is a genetic hotspot for several neurological diseases. A mutation of VPS35, the core component of the retromer complex that also contains VPS26 and VPS29 (Fig. 1b), causes a familial form of PD (PARK17)[32,33]. VPS35 deficiency has also been linked to Alzheimer's[34,35]. Retromer, predominantly residing on the early endosomes (EEs), is a master conductor that sorts cargoes to the trans-Golgi network, cell surfaces, late endosome (LE), or lysosome (Lys)[36,37] (Fig. 1b). The degradative compartments, LE and Lys, both have an acidic lumen and express the common markers Lamp1 and Lamp2. However, LE has a unique multivesicular lumen that also expresses the specific marker CD63[38,39].

Here, we developed a conditional mouse line, Rod$^{\Delta Vps35}$, of which the *Vps35* gene was selectively deleted in rods. Longitudinal studies showed these mice developed early rod terminal loss well in advance of rod cell death. The mutant retinas exhibited prominent P-αSyn- and ubiquitin-positive LB-like inclusions, secondary to the impaired LE lumen waste clearance of VPS35-knockout (KO) rods. The P-αSyn-lipofuscin aggregates accumulated in the phagocytic microglia exhibit autofluorescence (AF), which can be detected by fundoscopy in real time. Mechanistically, we uncovered VPS35's protein network through proteomics and its interaction with HSC70. VPS35-deficiency-caused aberrant LE sequestration of HSC70 and its aggregation with P-αSyn, providing keen insights into VPS35-deficiency-caused αSyn pathology.

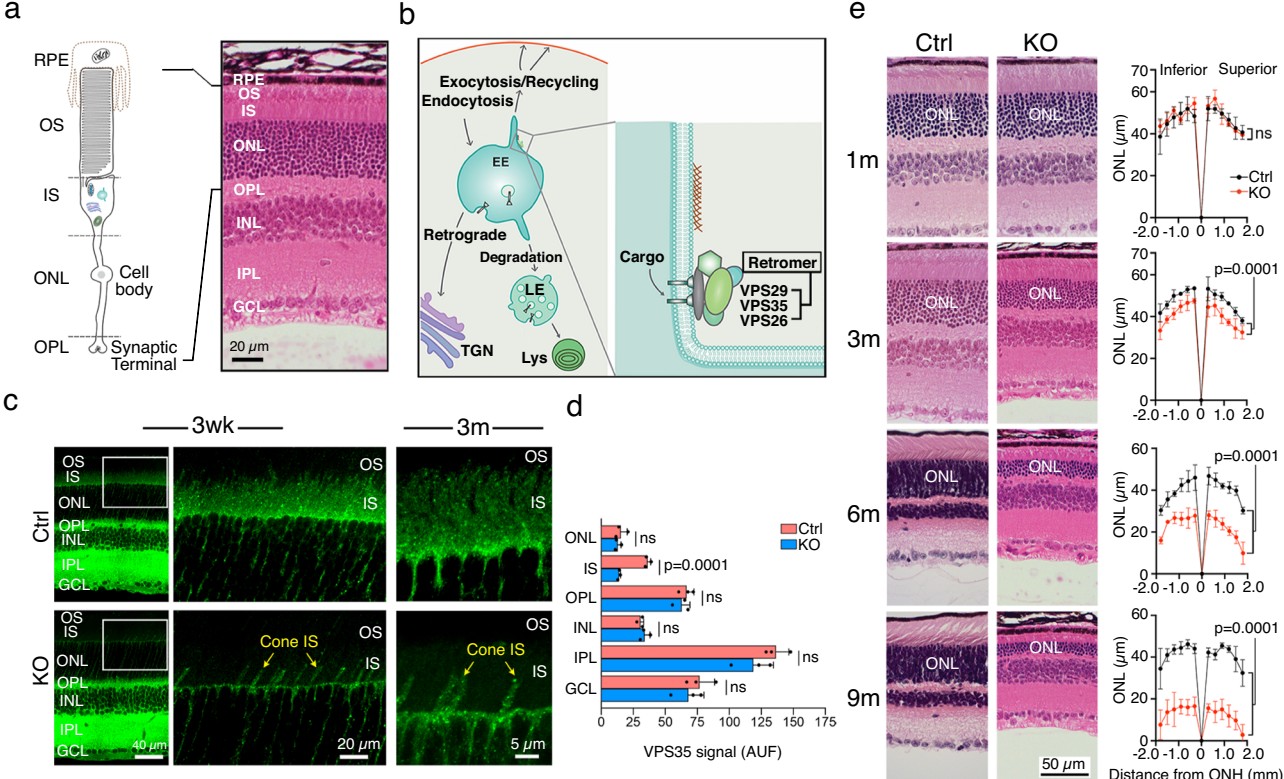

**Fig. 1 | Diagrams of rod and retromer-regulated trafficking as well as rod-loss and retinal degeneration in Rod$^{\Delta Vps35}$ mice. a** A diagram depicts the four compartments of a mammalian rod and their respective retinal layer distribution shown in the HE-stained mouse retinal section. **b** A diagram illustrates the multiple retromer-controlled vesicular trafficking pathways. The VPS26-VPS29-VPS35 retromer complex primarily resides in the junction between the globular and tubular domains of the EE. TGN: Trans-Golgi network. **c** VPS35 immunostaining in 3-week (wk)-old and 3-month (m)-old Ctrl and KO mouse retinas. Enlarged views show that VPS35 puncta normally observed in the IS region of Ctrl is diminished in KO at both ages. The residual VPS35 signal is derived from the IS of the morphologically

characteristic cones. **d** Quantification of VPS35 staining signal in different retinal layers of 3-month-olds. AUF arbitrary unit of fluorescence. Mean ± SD and *P*-values of *N* = 3 in each group are shown. Two-tailed Student's *t*-test. **e** (Left) Representative age-matched H&E-stained mouse retinal sections. (Right) measurement of the ONL thickness across the superior and inferior retinal hemispheres in relation to the distances (μm) to the optical nerve head (ONH; considered as 0). Mean ± SD and *P*-values of *N* = 3 in each group are shown. Two-way ANOVA test. Representative data of 3 independent repeats of (**c**, **e**) are shown. Source data of (**d**, **e**) are provided as a Source Data file.

## Results

### VPS35 deficiency causes early synapse loss preceding rod cell death

The rod-specific VPS35-KO mouse line (Rod$^{\Delta Vps35}$,) was generated by crossing a rod-specific Cre mouse line (iCre75) and a Vps35-floxed mouse line (VPS35$^{f/f}$). Age-matched iCre75$^{+/-}$; VPS35 $^{f/f}$ (herein KO) and iCre75$^{-/-}$; VPS35$^{f/f}$ (herein Ctrl) littermates were analyzed in parallel. The Cre in iCre75 mice was shown to be activated at postnatal 4–7 days[40] and its rod-specificity has been extensively characterized[40–43]. We too confirmed the rod-specific Cre expression based on its ONL-restricted staining (Supplementary Fig. 1a). As rods contribute to the majority of total photoreceptors[44], the VPS35 puncta staining typically observed in the IS layer of Ctrl mice was largely absent in both 3-week-old and 3-month-old KO (Fig. 1c, d). The co-staining with ATP1A (Na$^+$, K$^+$-ATPase), which marks the photoreceptor IS plasma membrane, further verified the IS-expressed VPS35 was largely diminished in KO (Supplementary Fig. 1b). The co-staining with cone arrestin, that marks cone, showed the residual VPS35 in the IS region was from cones (Supplementary Fig. 1c). VPS35 puncta in ONL were largely present in glutamine synthase-labeled Müller glial processes spanning across ONL (arrows in Supplementary Fig. 1c). The Müller glial process associated VPS35 signals were indistinguishable between Ctrl and KO (Supplementary Fig. 1d). Thus, the total VPS35 signals in ONL were unchanged in KO (Fig. 1d). The overall VPS35 signal in the OPL was also unchanged (Fig. 1d), probably because the OPL not only contains rod terminals but also the terminals of several other neuron cell types (cones, bipolar cells, and horizontal cells) as well as the lateral processes of Müller glia. To specifically test the rod terminal loss of VPS35 in KO, we performed co-staining with Ribeye, which marks the rod synaptic ribbon where the SVs dock (Supplementary Fig. 1e). In contrast to the Ctrl where VPS35 was often associated with Ribeye, the KO had significantly decreased colocalization between VPS35 and Ribeye (Supplementary Fig. 1f). The VPS35 staining in the rest of the retinal layers (i.e., inner nuclear layer (INL), inner plexiform layer (IPL), ganglion cell layer (GCL)) remained unchanged both qualitatively (Fig. 1c) and quantitively (Fig. 1d) in KO. These results comprehensively validate the rod-specific loss of VPS35 in Rod$^{\Delta Vps35}$ mice.

The hematoxylin and eosin (HE)-stained histological sections (Fig. 1e) and SD-OCT fundus scans of live mice (Supplementary Fig. 1b, c) both revealed progressive retinal degeneration of the KO. The thinning of the ONL became significant, beginning at 3 months of age, and worsened over time.

We used an electroretinogram (ERG) to measure the electrical activity of the retina in response to light. The scotopic (rod) ERG signals, both a- and b-wave, were significantly reduced in 1-month-old KO (Fig. 2a). The b-wave of the photopic (cone) ERG signal was also reduced, albeit to a lesser degree (Fig. 2b). Although there were still a few rows of photoreceptor nuclei remaining in 9-month-old KO (Fig. 1e), these mice had no detectable scotopic and photopic ERG signals (Fig. 2a, b). These results suggest that rod terminals undergo functional decline much before the eventual rod cell death in Rod$^{\Delta Vps35}$ mice.

The early anomaly of the rod terminals was further indicated by the reduced staining of Ribeye, and SNAP25, the marker of the synaptic plasma membrane, in 1-month-old KO (Fig. 2c). The density of the synaptic contacts between rods and bipolar cells, measured by metabotropic glutamate receptor 6 (mGluR6), which is the post-synaptic terminal marker, was also reduced (Fig. 2d). Transmission EM showed that remaining rod terminals in 1-month-old KO feature a deflated appearance and dark endosomal tubules which were not observed in Ctrl (Fig. 2e).

### VPS35-KO rods have dysregulated OS genesis, endolysosomal homeostasis, protein degradation, and autophagic flux

The measurement of the OS in HE-stained sections showed it become significantly shorter in 3-month-old KO (Fig. 1e, Supplementary

Fig. 2a). However, its ultrastructure already looked abnormal in 1-month-old by EM. Unlike the Ctrl OS densely packed uniform-shaped disc membranes (Fig. 3a), the KO had loosely packed disc membranes interspersed with short tubules and EE-like vacuoles (Fig. 3b). Quantitative immunoblotting assays (Fig. 3c) and immunostaining (Supplementary Fig. 2b) and revealed the steady-state expression of all OS proteins tested, including peripherin2 (PRPH2), rhodopsin, phosphodiesterase 6 (PDE6), ATP Binding Cassette Subfamily A Member 4 (ABCA4), and Interphotoreceptor retinoid-binding protein (IRBP) were significantly reduced. However, qPCR assays showed that the mRNA levels of these molecules were unchanged (Supplementary Fig. 2c). These observations indicate that these proteins are proteolytically degraded due to their faulty OS localization.

VPS35 perturbations causing EEs collapse[45] and autophagy dysregulation[46–48] have been previously reported in cell culture studies. We measured the plausible change caused by VPS35 deficiency in the endolysosomal-autophagic homeostasis in rods in vivo by quantitative immunostaining of the organelle-specific markers, EM, and biochemical assays. In KO, the EEA1-labeled EEs localized in the IS were largely scarce (Fig. 3d, e). EM showed the residual EEA1 signals observed by light microscopy represented unusual clusters of EEs collapsed at the IS-ONL border (Fig. 3d, inset). In contrast, Lamp1-labeled LEs/Lys, CD63-labeled LEs, and LC3-labeled autophagic vacuoles were significantly increased in the IS (Fig. 3d, e) and OPL (Fig. 3d, f) regions of KO. The higher LC3 signals indicated the activation of macroautophagy flux, which was supported by the increase in lipidated LC3II, but not soluble LC3I, in KO (vs. Ctrl) on immunoblots (Fig. 3g). Nonetheless, the KO-expressed LC3- and Lamp1-labeled puncta KO retina did not overlap well (Pearson's coefficient = 0.37 + 0.08; Fig. 3h), indicating impaired fusion of autophagy and lysosome (i.e., autophagolysosome maturation).

### Lipofuscins deposits are undigestible LE luminal aggregates

Using super high Z-resolution (10 nm/pixel Z-section) 3D-focused ion-beam scanning EM (FIB-SEM), we confirmed that, as previously reported[49], normal mouse rods exhibit some morphologically typical LEs but no detectable Lys. Consistent with the increased CD63 staining, FIB-SEM observed an expansion of enlarged multivesicular LEs in the IS (Fig. 4a, b, Supplementary Movie 1) and OPL (Fig. 4c–e) of 3-month-old KO. The LE lumen in the VPS35-KO rod terminals was abnormally filled by vesicles identical to the SVs in the synaptic cytoplasm (Fig. 4c, Supplementary Movie 2), an indication of impaired SV recycling.

Inspection of the stacked images depicted the intraluminal vesicles retained in the LE lumens were transformed into amorphous aggregates (Fig. 4d, e; Supplementary Movies 1–4), which were then turned into electron-dense pigment deposits (aka lipofuscins) in the cytoplasm. The concentric membrane whorls, aka multilaminar bodies (MLBs), derived from the autophagosomes that failed to form autophagolysosomes[50,51], were also abundant in the VPS35-KO rod terminals (Fig. 4d, f, g; Supplementary Fig. 3b, c). In contrast, Ctrl rod terminals exhibit almost no detectable SVs- or amorphous aggregates-filled LEs, lipofuscins, and MLBs (Supplementary Fig. 3a).

The FIB-SEM survey readily visualized the escape of lipofuscins, MLBs, and other membrane debris in bulk from KO rod terminals (Fig. 4f, Supplementary Movies 1, 4), their appearance in the extracellular space, and their emergence in neighboring glial cells (i.e., microglia, Müller glia) probably through phagocytosis (Fig. 4g, Supplementary Fig. 3b, c; Supplementary Movie 4; see later Fig. 7e). The counting studies revealed microglia exhibited more lipofuscins (3.98 counts/mm$^3$) than MLBs (1.62 counts/mm$^3$), whereas Müller glia exhibited more MLBs (14.46 counts/mm$^3$) than lipofuscins (2.35 counts/mm$^3$) (Supplementary Fig. 3b, c). These ultrastructural studies visually portray the natural history of the LE lumen accumulated aggregates. They also provide morphological evidence depicting the cell-to-cell transmission of these aggregates from VPS35-deficient nerve terminals.

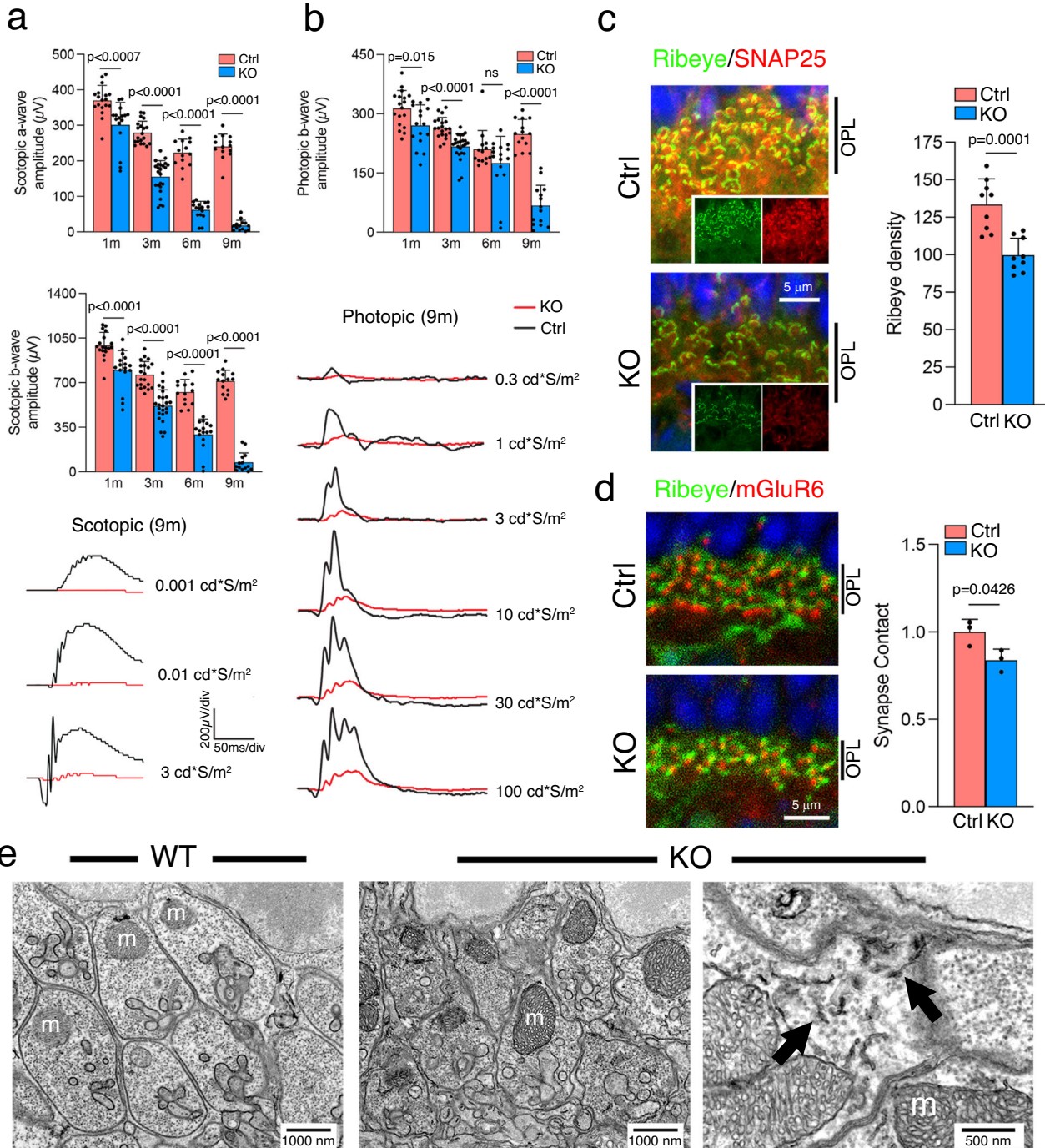

**Fig. 2 | Early structural-functional decline of VPS35-KO rod terminals.**
**a**, **b** Representative scotopic (**a**) and photopic (**b**) ERG amplitudes, evoked by 3 cd*s/m² and 100 cd*s/m² flashes, respectively, show the age-dependent ERG decline. Means ± SD are shown. N = 18, 20, 14, and 14 eyes for 1-, 3-, 6- and 9-month-old Ctrl mice, respectively, and N = 16, 24, 16, and 14 eyes for 1-, 3-, 6- and 9-month-old KO mice, respectively. Representative traces, evoked by a range of light intensity, of 9-month-old Ctrl and KO (N = 14 eyes for both Ctrl and KO eyes) are shown. Mean ± SD and P-values are shown. Two-tailed Student's t-test. **c**, **d** OPL staining shows the merged and single channel views (insets) of Ribeye, SNAP25, and DAPI (**c**) or Ribeye, mGluR6 and DAPI (**d**) in 1-month-old mice. Representative figures of 3 independent repeats are shown. The Ribeye signal density (puncta/1000 μm²) of KO and Ctrl (**c**) and the fold change of mGluR6-labeled synaptic contacts in KO (vs. Ctrl as 1) (**d**) are also shown. For (**c**, **d**), Mean ± SD and P-values of N = 3 mice are shown. Two-tailed Student's t-test (**c**, **d**). Source data of (**a–d**) are provided in Source Data file. **e** Electron micrographs of rod terminals in 1-month-old wild-type (WT) and KO. Open arrows mark the aberrant short, dark tubules observed in the KO terminals. m mitochondria.

## Rod$^{\Delta Vps35}$ mouse retinas develop hallmarks of synucleinopathy

VPS35 has a reported role in preventing αSyn aggregation through mechanisms that are incompletely understood[52–54]. We observed that the difference in αSyn expression between Ctrl and KO was through the appearance of αSyn-labeled granules in the IS-OS region of KO (Supplementary Fig. 4a). In contrast, P-αSyn staining was significantly increased in all retinal layers in 3-month-old KO (Fig. 5a, b), indicating its propagation and spreading to the inner retinas. The inner retinal pathology of the KO was also associated with the declined visual acuity (Supplementary Fig. 4b).

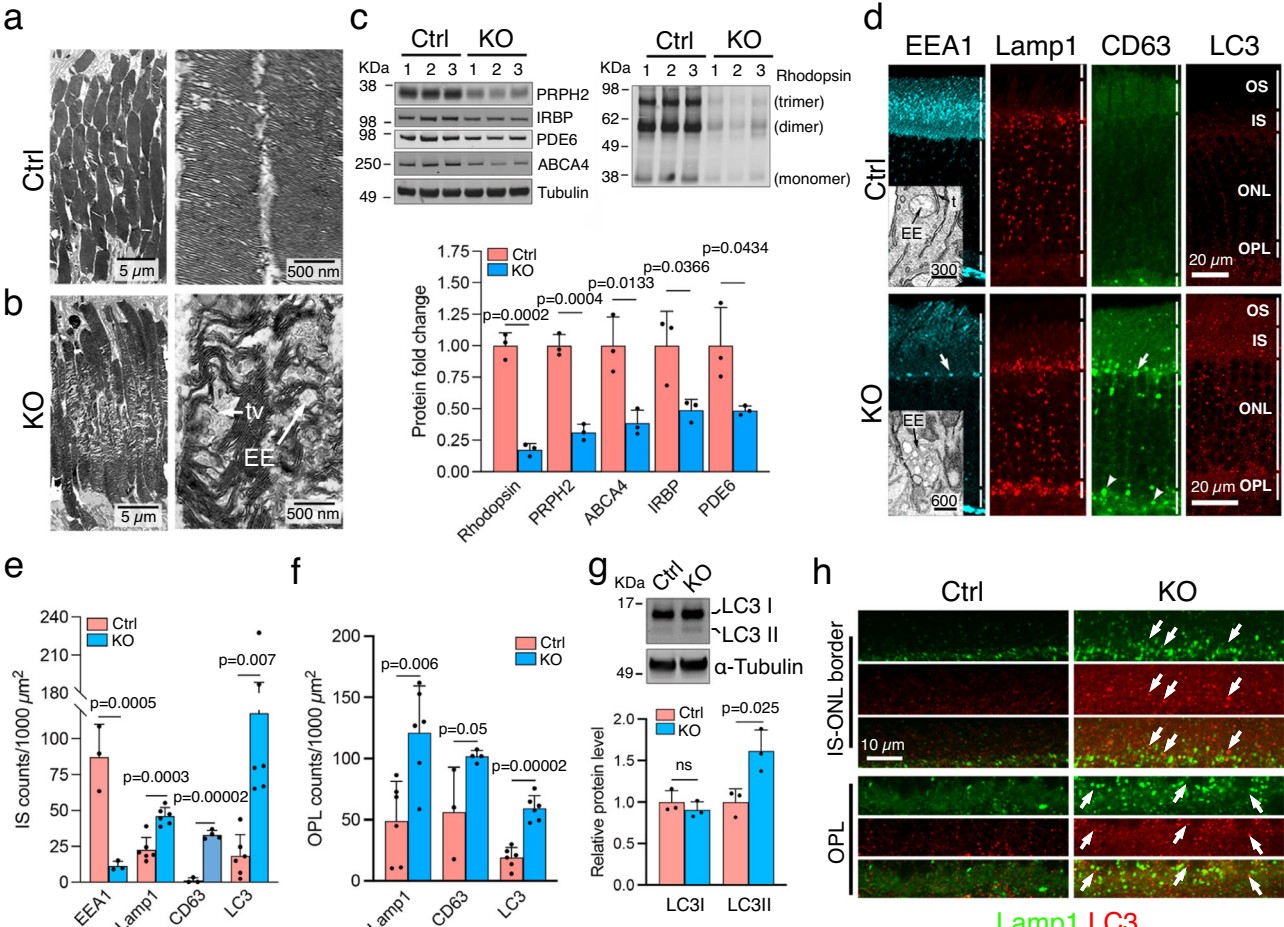

**Fig. 3 | VPS35 deficiency dysregulates OS morphology, OS protein turnover, and endomembrane homeostasis in rods. a, b** Low (left) and high (right) magnifications of electron micrographs showed the OS in 1-month-old Ctrl and KO mice. tv: tubulovesicles. **c** Immunoblots of indicated OS proteins. As expected, rhodopsin has a propensity to form monomers, dimers, and trimers. Bar graphs show the relative intensity of protein bands of indicated molecules (normalized using α-tubulin as an input control) considering Ctrl as 1. **d** 3-month-old mouse retinas stained by indicated markers. Arrows mark the residual EEA1 signals and increased CD63-labeled LEs at the IS-ONL borders. Arrowheads mark the increased OPL expression of CD63-labeled LEs. Insets: (Top) Electron micrograph shows Ctrl rod IS has solitary EE globes surrounded by tubules (t). (Bottom) EE clusters are collapsed in the lower part of the KO rod IS. Bars in nm. **e, f** Bar graphs show the fold

change in signal density (counts per 1,1000 mm²) of indicated markers in IS (**e**) and OPL (**f**). For EEA1, $N = 3$ mice each for both Ctrl and KO (1 image per animal); for CD63, $N = 3$ mice for Ctrl and $N = 4$ mice for KO animals (1 image per animal); for Lamp1 and LC3, $N = 3$ mice each for both Ctrl and KO (two images per animal) are shown. **g** LC3 immunoblots of in 3-month-old mouse retinas and intensity of the LC3I and LC3II signal (normalized using α-tubulin as an input control) considering Ctrl as 1. **h** Co-staining of Lamp1 and LC3 in the indicated retinal layers of 3-month-olds. Arrows mark LC3-postive, Lamp1-negative/ puncta. For **c**, **e**–**g**, Mean ± SD and $P$-values of $N = 3$ in each group are shown. Two-tailed Student's $t$-test. Representative images of 3 independent repeats (**a**–**e**, **g**, **h**) are shown. Source data of (**c**, **e**–**g**) are provided as a Source Data file.

Most notably, inclusions positively labeled with P-αSyn and ubiquitin were conspicuous in the photoreceptor layers in the KO, but not Ctrl (Fig. 5c). These inclusions were also highly positive for CD63 (Fig. 5c), suggesting they encompass the LE luminal elements. Recent FIB-SEM studies have shown that human LBs inclusions encircle a mixture of lipofuscins and membranous/organelle debris[55]. Using immunoEM, we showed that inclusion bodies shared striking similarities to LBs were found in the OPL and IS-OS regions of 3-month-old KO and these inclusions were labeled with P-αSyn (silver-enhanced immunogolds) (Fig. 5d).

αSyn within inclusions has been shown biochemically to be resistant to extraction with Triton X-100 (TX-100)[56]. Our pilot studies showed that the major retinal protein rhodopsin interfered with the detection of αSyn, most likely due to technical reasons (e.g., rhodopsin has a concentration- and heat-dependent oligomerization, see Fig. 3c). Since αSyn is not expressed in the OS (Supplementary Fig. 4a), we removed OS from the retinal homogenates prior to TX-100 extraction (see Methods). The insoluble pellets were subjected to 8 M urea + 2%

SDS extraction. Immunoblotting assays showed that αSyn was abundantly expressed in the soluble fractions; it can be detected in as little as 0.1 mg of OS-depleted retinal lysates. It migrated around the predicted size of tetramers and dimers (Fig. 5e). This migration pattern was largely unchanged under different denaturing conditions tested (Supplementary Fig. 4c). While P-αSyn was undetectable in the soluble fraction, it was detected in the TX-100 insoluble fraction as monomers, truncated monomers, and various-size oligomers (Fig. 5e). Quantification showed the KO (vs. Ctrl) expressed more of both soluble αSyn and insoluble P-αSyn (Fig. 5f). The results, using multiple approaches, consistently suggested that Rod$^{ΔVps35}$ mouse retinas harbor αSyn aggregates and LB-like inclusions.

**Proteomics and pathway analyses of VPS35-interacting proteins**
To uncover the mechanism(s) underlying VPS35-deficiency-caused P-αSyn aggregation, we searched for VPS35-interacting proteins. We established an immunoprecipitation (IP) and elution protocol that specifically pulled down the endogenous VPS35 from C57BL6/J mouse

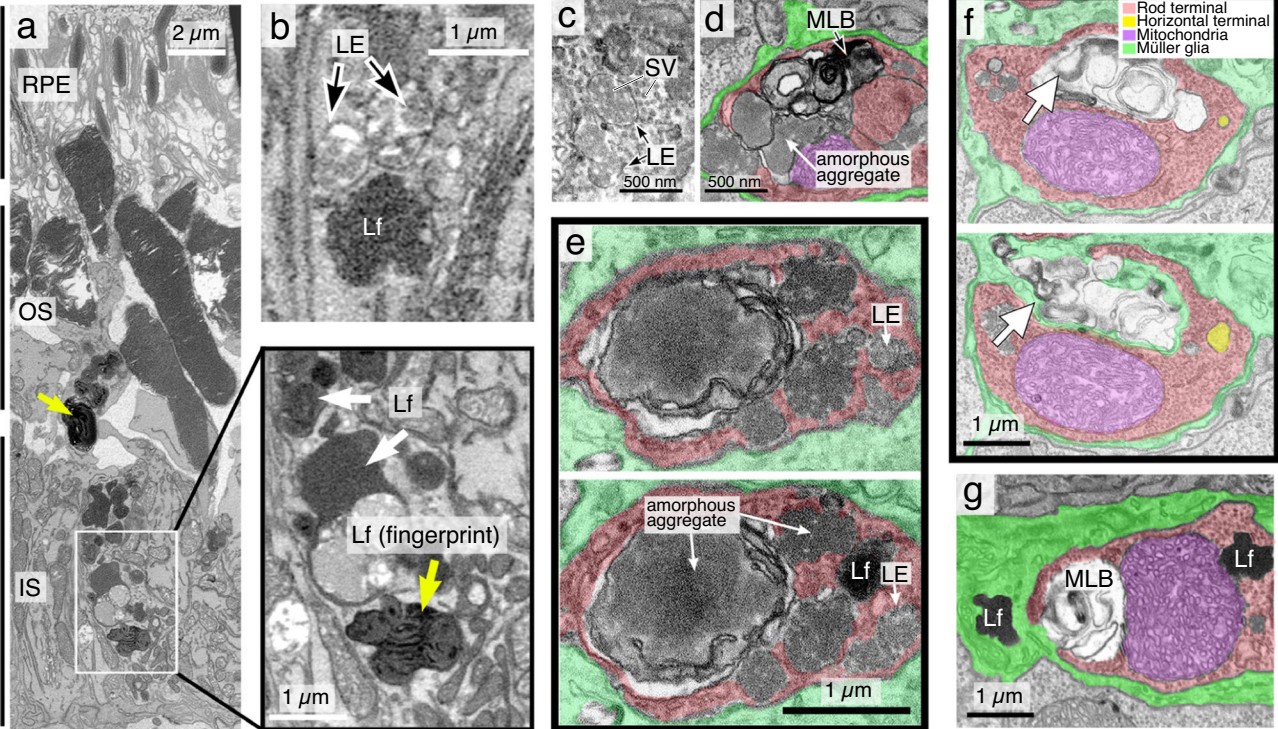

**Fig. 4 | VPS35-KO rods feature impaired clearance and SV recycling. a–g** FIB-SIM images of rod IS (**a**, **b**) and synaptic terminals (**c–g**) of ~3-month-old KO mice. The colors that mark different subcellular structures are labeled in (**f**). Yellow arrows mark the lipofuscins had a fingerprint pattern (**a**). The LEs in the rod terminals were packed with SVs (**c**) or amorphous aggregates (**d**, **e**). The two consecutive images in (**e**) highlight the deposition of LE luminal wastes as lipofuscins (also see Movie 1–4). The two consecutive images (**f**) highlight the bulk release of lipofuscins, MLBs, and various membrane organelles/debris from a rod terminal. **g** shows two similar-looking lipofuscins, one inside and another outside rod terminal. Lf lipofuscins. Representative images of two independent studies are shown.

retinal lysates (Supplementary Fig. 5a). VPS26 was co-immunoprecipitated, as expected, validating the IP procedures. We subjected the VPS35-IP samples to the LC–MS/MS analyses.

The VPS35-IP proteomics data showed an accumulative of 211 molecules from three rounds of experiments (Supplementary Table 1). Bioinformatics analyses revealed 36 significant KEGG pathways from DAVID analyses (Fig. 6a, adjust *p*-value < 0.05, Supplementary Table 2), 63 significant GO pathways by the Fisher's test (Fig. 6b; adjusted *p*-value < 0.5, Supplementary Table 3), and 41 significant GO pathways from the GSEA method (FDR < 0.2; Supplementary Table 4). The latter two method sets shared 12 common significant pathways (Fig. 6c). Some of the overrepresented pathways are linked with previously implicated functions of VPS35 such as intracellular trafficking, PD, Alzheimer's, and SV cycling (Fig. 6a, b), which validated the IP proteomics experiments. Our analyses also highlighted several pathways that have not been previously recognized for VPS35. These include protein processing in the endoplasmic reticulum (ER), protein refolding, proteosomes, cell junctions, cytoskeleton constituents, polymeric cytoskeletal fibers, supramolecular polymers, and many more.

### Characterizing the physical and functional HSC70-VPS35 interaction

We focused on studying HSC70, one of the VPS35-interacted proteins, because of its implication in αSyn folding and PD pathology[24,57–59]. The list of VPS35 immunoprecipitants was also enriched for several known HSC70 client proteins (i.e., Histone H3.1, glyceraldehyde 3-phosphate dehydrogenase[60], pyruvate kinase, alpha-enolase, ATP5A1, Slc25a5, Elongation factor 1-alpha 1, polyubiquitin-C[61–63]), and HSC70 binding proteins (i.e., alpha-crystallin, ezrin[64]). We confirmed the physical interaction between VPS35 and HSC70 using pull-down assays. These studies showed that His-tagged HSC70 bound to glutathione S-transferase (GST)-tagged VPS35, but not to GST alone (Fig. 6d). Further characterization revealed that the C-terminal half of VPS35 largely constituted its interaction with HSC70, rather than the N-terminal half of VPS35 (Fig. 6d).

A previous study has shown that HSC70 is enriched in the endosomes of biochemically fractionated HeLa cells[65]. Consistent with this report and our pull-down assays, we found that HSC70 colocalized with VPS35 (Fig. 6e), EEA1, and Lamp1 (Fig. 6f) in the photoreceptor-like cell line 661W. The staining of HSC70 is specific because it disappeared upon suppressing HSC70 through transfection with HSC70-short hairpin RNA (shRNA) (Supplementary Fig. 5b). Like in 661W cells, HSC70 and VPS35 signals were also overlapped in rods isolated from Ctrl mouse retinas (Supplementary Fig. 5d). In contrast, in KO rods HSC70 became enriched in CD63-labeled LEs that also contained αSyn signal (Supplementary Fig. 5e). Staining of 3-month-old retinas also confirmed the prominent appearance of HSC70 in CD63-labeled LEs (Supplementary Fig. 5f, g).

We subsequently investigated the possibility that VPS35 deficiency results in LE sequestering of HSC70 and αSyn in 661W cells. Transfection alone with Myc-tagged αSyn resulted in a weak and diffused signal, and it did not alter the expression pattern of endogenous HSC70 (Fig. 6g). In contrast, co-transfection with VPS35-shRNAs (which were previously reported[66] and we too experimentally confirmed; Supplementary Fig. 5c), caused a significant increase in large Myc-αSyn+/HSC70+ aggregates in CD63-labeled LE lumens (Fig. 6g) and Lamp1-labeled vacuoles (Fig. 6h). The Myc-αSyn aggregates in VPS35-suppressed cells also exhibited P-αSyn (Fig. 6h). These results collectively suggest VPS35 deficiency promotes the sequestering of HSC70 in the LE lumen where it forms aggregates with the presumably unfolded/misfolded αSyn and accumulates P- αSyn. This is underlined by the fact that HSC70 is an LB component in PD patient brains[67].

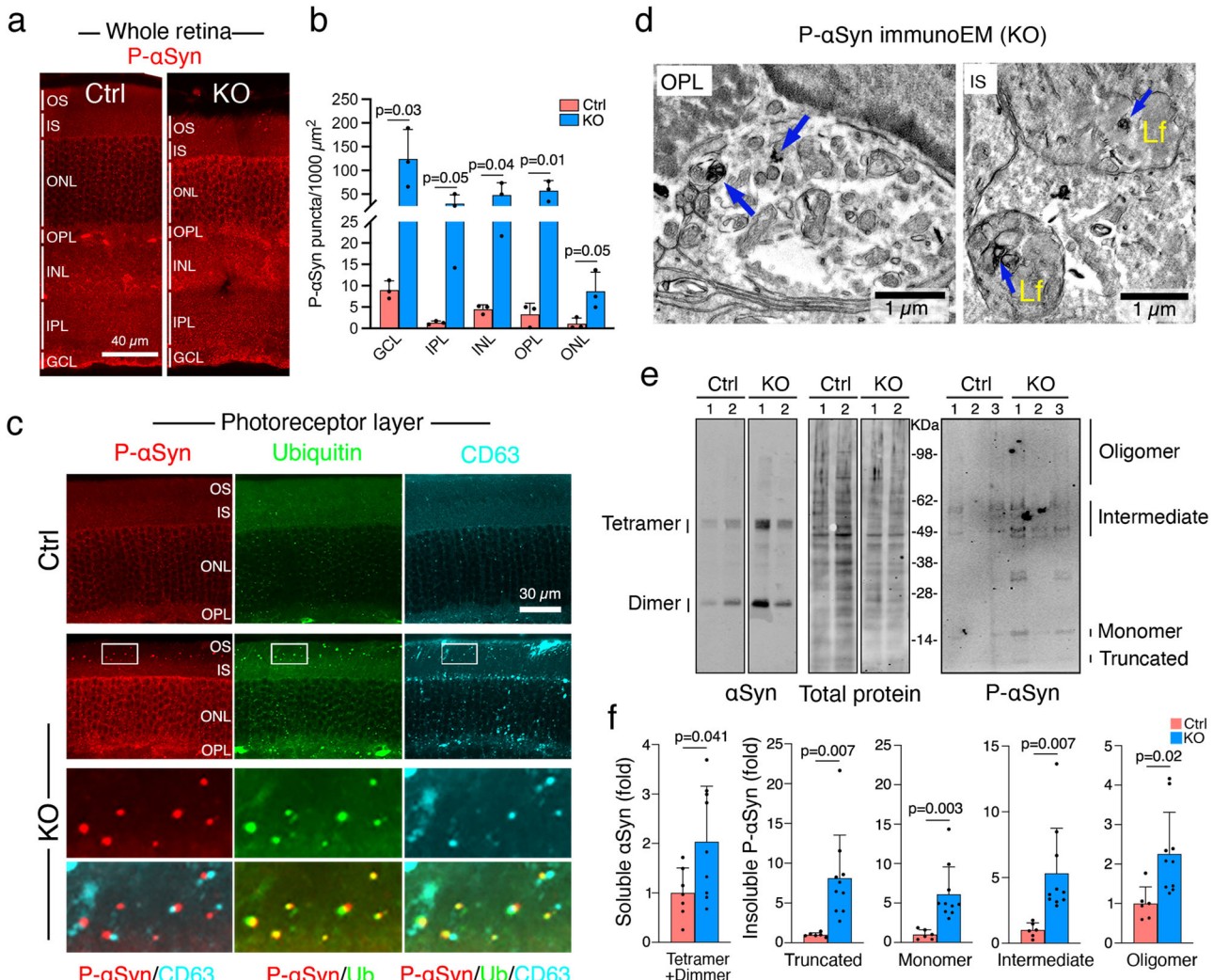

**Fig. 5 | Rod^ΔVps35 mouse retinas exhibited P-αSyn aggregates and LB-like inclusions. a**, **b** Whole retinal views of P-αSyn staining in 3-month-olds (**a**). Bar graphs show the P-αSyn puncta counts (per 1000 mm²) of different retinal layers (**b**). $N = 3$ for each group. Mean ± SD and *P*-values. Two-tailed Student's *t*-test. **c** Confocal images of P-αSyn, ubiquitin (Ub), and CD63-stained photoreceptor layers in 3-month-old mouse retinas. Triple positive LB-like inclusions distributed in the IS-OS region of KO are shown in the enlarged boxed areas. **d** ImmunoEM images show that the OPL and IS regions (of 3-month-old KO) exhibited sliver-enhanced P-αSyn immunogolds in LB-like inclusion (blue arrows, left) and are associated with lipofuscins (blue arrows, right). Lf: lipofuscins. **e** Representative immunoblots of αSyn and total protein stain of TX-100-soluble retinal extracts (Left). Immunoblots of P-αSyn in TX-100 insoluble/SDS-urea soluble retinal fractions (Right). **f** (Left) The quantification of soluble αSyn (combined tetramer and dimer signals, normalized using total protein as an input control, considering Ctrl as 1; $N = 9$ and 7 for KO and Ctrl, respectively). (Right) The quantification of differentially migrated P-αSyn bands of insoluble fractions ($N = 6$ and 10 for KO and Ctrl, respectively). Mean ± SD and *P*-values. Two-tailed Student's *t*-test. Representative images of 3 and 2 independent repeats of (**a**–**c**, **e**) and (**d**), respectively, are shown. Source data of (**b**, **f**) are provided as a Source Data file.

## Microglia engulfed with P-αSyn inclusions are pathological and exhibit bright AF

We tested the plausible microglia activation in response to the rod deficiency of VPS35. Iba1-labeled microglia were exclusively restricted in the OPL and IPL in Ctrl (Fig. 7a), as expected. In KO, microglia often migrated to the IS-OS region and accumulated in the subretinal region (between OS and RPE) (Fig. 7a). To characterize the activation state of the microglia distributed in different layers, we performed morphometric analyses of the *en face* views of the microglia in retinal flat mounts. The measurement of their perimeter and radius showed that in 3-month-old KO, microglia in the subretinal region had less ramification than those in the OPL and IPL (Fig. 7b, Supplementary Fig. 6a). And the OPL- and IPL-localized microglia in KO also had less ramification than their counterparts in Ctrl (Fig. 7b). Microglia with little ramification were also found in the subretinal space of 6-month-old KO but not Ctrl (Supplementary

Fig. 6b). Furthermore, the subretinal (vs. OPL) microglia expressed more inflammatory cytokine IL-1β and CD68, a microglia activation marker (Supplementary Fig. 6a, c). Albeit weaker, the level of IL-1β and CD68 expressed by OPL microglia of KO (vs. Ctrl) was significantly higher (Supplementary Fig. 6a, d).

The above results collectively suggest that, while the microglia were overall more activated in the KO, the subretinal infiltrated microglia are the most active ones. Consistently, the subretinal microglia also displayed bright staining of the disease-associate microglia (DAM) marker[68,69]- Triggering Receptor Expressed on Myeloid Cells 2 (TREM2) in conjunction with galectin3 (Fig. 7c), and, importantly P-αSyn (Fig. 7d). Since microglia did not typically express αSyn (Supplementary Fig. 4a), the P-αSyn detected in the subretinal microglia likely conferred the engulfment of VPS35-KO rod terminals deposited aggregates. This hypothesis was supported by the FIB-SEM inspection that showed the microglia in both OPL (Fig. 7e) and

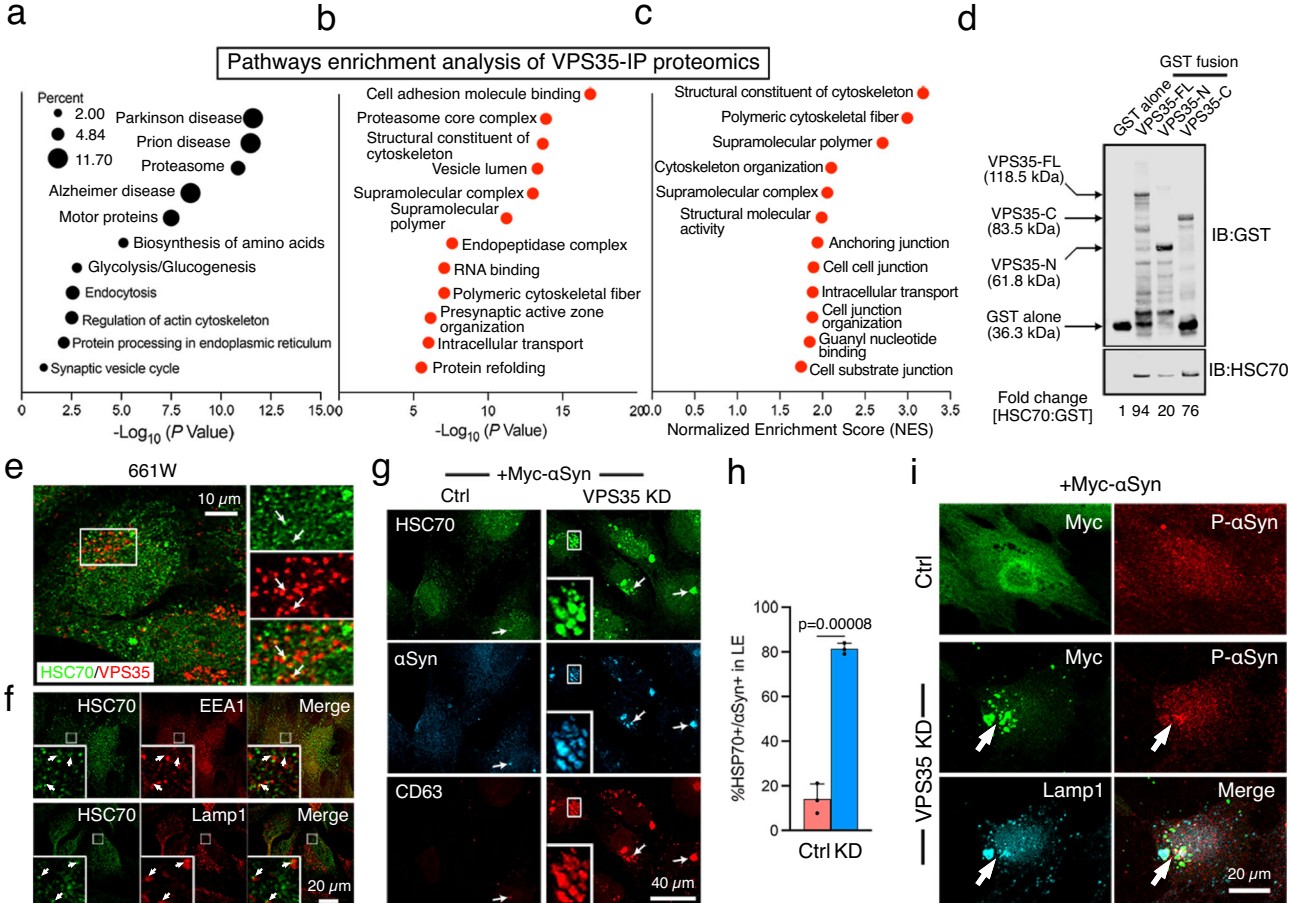

**Fig. 6 | Proteomics pathway analyses of VPS35-interacted proteins and VPS35-HSC70 interaction in α-Syn aggregation. a** Pathway analyses of the proteomics dataset of VPS35 immunoprecipitants (IP) from mouse retinal lysates. Selected enriched pathways of VPS35-interacting proteins analyzed by DAVID are shown in (**a**). The -Log$_{10}$ of *P*-values from one-sided Fisher's exact test are shown. **b, c** The enriched Gene Ontology (GO) pathways were identified by one-sided Fisher's test and the GSEA method. The *P*-values from the Fisher's test were adjusted by using Bonferroni correction. There are 63 significant GO pathways with adjusted *P*-value < 0.05 from Fisher's test and 41 significant GO pathways with FDR < 0.2 from GSEA. Twelve pathways were shared by the two significant GO pathway sets obtained from Fisher's test and GSEA. For these pathways, the -Log$_{10}$ of *P*-values from Fisher's test are shown in (**b**) and the normalized enrichment scores from GSEA are plotted in (**c**). **d** Pull-down assays. GST alone or GST-VPS35 fusions conjugated on glutathione beads were mixed with HSC70. The glutathione elutes were immunoblotted (IB) with GST and HSC70. The expected size of GST fusions of

VPS35 full-length (FL), C-terminal fragment (C), and N-terminal fragment (N) are shown. The relative ratio of the signal intensity of HSC70: GST (GST alone of GST-VPS35 fusions), considering GST alone as 1, are shown. **e, f** Immunostaining of HSC70 together with VPS35 (**e**) or EEA1 or Lamp1 (**f**) in 661 W cells. The overlapped signals are highlighted by arrows in insets. **g–i** 661W cells expressed plasmids encoding Rab5Q79L, Myc-αSyn, and pRK5 (Ctrl) or VPS35-shRNAs (KD) for 3 days and stained for indicated antibodies. **g** Representative images show VPS35-KD cells expressed prominent HSC70$^+$/αSyn$^+$ aggregates in CD63-labeled LEs (arrows). **h** Quantification shows significantly more VPS35-KD cells exhibited αSyn$^+$/HSC70$^+$ aggregates in Lamp1-labeled LEs. More than 200 transfected cells per condition were counted in 3 independent assays. Mean ± SD and *P*-values. Two-way Student's *t*-test. **i** VPS35-KD cells expressed prominent Myc-αSyn$^+$, P-αSyn$^+$ aggregates in Lamp1+ LEs (arrows). Ctrl cells expressed diffused Myc-αSyn and P-αSyn signals instead. Representative images of 3 independent images of (**d–i**) are shown. Source data of (**h**) provided as a Source Data file.

subretinal (Fig. 7f) regions of KO were heavily laden with lipofuscins resembling those observed in the neighboring rod terminals.

In agreement with the AF property of lipofuscins[64], almost all Iba1 and galectin-3-labeled subretinal microglia exhibited bright AF granules (Fig. 7f). This was best visualized by the *en face* views of the subretinal microglia adhered to the apical side of RPE upon retinal peeling. While the AF signal did not overlap well with galectin-3 (Fig. 7f), the AF signal had an extensive colocalization with P-αSyn staining (Fig. 7g; Pearson's coefficient 0.603 ± 0.078 and 0.475 ± 0.073 for 3-month-old and 7-month-old KO. *N* = 6 and *N* = 19 imaging fields for 3- and 7-month-old, respectively). The P-αSyn in the subretinal microglia was resistant to protease K treatment (Fig. 7h), a feature characterized by the misfolded αSyn in synucleinopathy[70,71]. The Pearson's coefficients showed that overlap between P-αSyn and AF was unchanged by protease K treatment (Fig. 7i). Of note, we used Alexa 647 dye-conjugated secondary antibody to visualize P-αSyn labeling as lambda scans

showed that the fluorescence of the microglial expressed AF did not overlap with the 653 nm/emission 668 nm barrier filter. To independently validate the close affiliation between P-αSyn and AF (and completely rule out the "bleeding" possibility), we used immunoEM. These results depicted the P-αSyn labeling of lipofuscins, which were often inside the LB-like inclusions accumulated at the subretinal space (Fig. 7j).

**Fundus AF as a surrogate of retinal synucleinopathy and atrophy**

The histological studies described above suggest that the AF concentrated in the microglia is a good surrogate of the pathological αSyn lesion. In live KO mice, the subretinal microglia-corresponding white dots, detected by color fundus photographs, completely coincided with the dots seen by fluorescence fundoscopy (Fig. 8a). Similar white and fluorescent dots were not observed in Ctrl. The confocal scanning

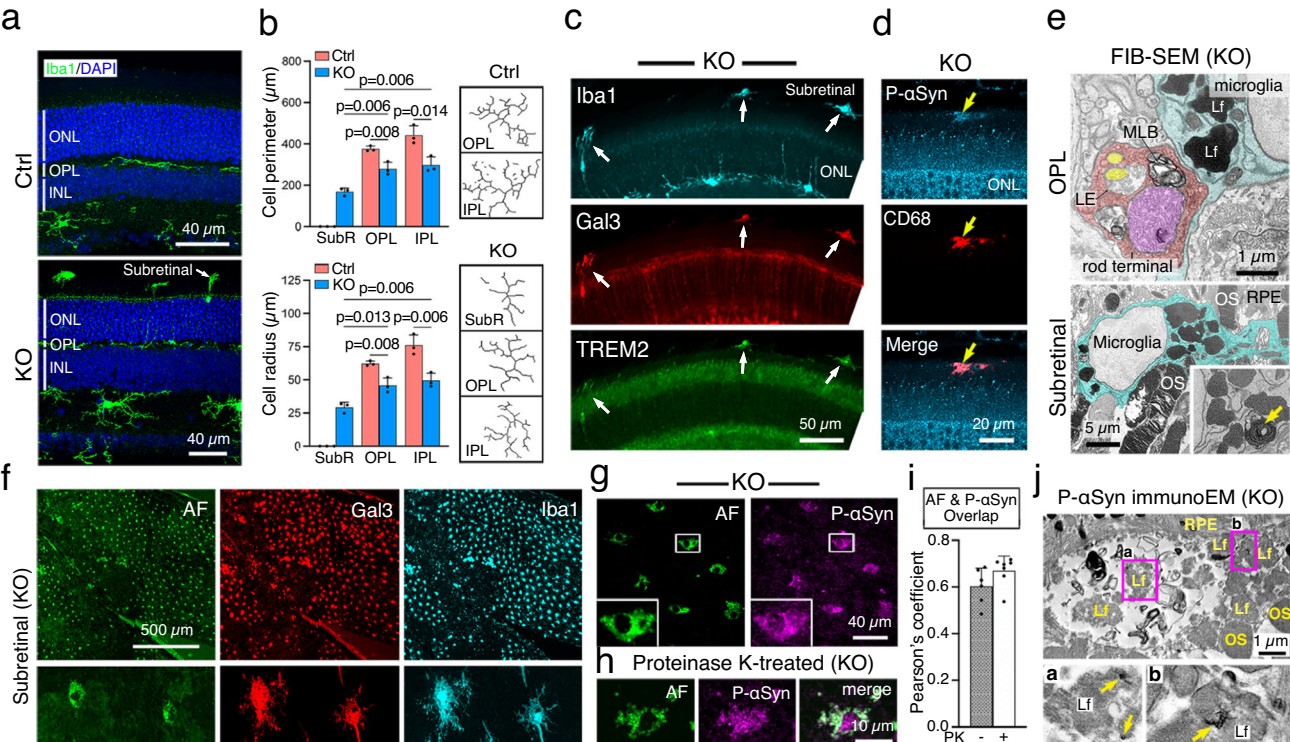

**Fig. 7 | VPS35 deficiency of rods activates microglia which infiltrate and display P-αSyn-decorated AF lipofuscins. a** Whole retinal sectional views of Iba1 staining of 3-month-olds. **b** Representative skeletons of Iba1-labeled microglia distributed in different retinal layers of 3-month-old flat mounts as well as the quantification of their average perimeter (top) and radius (bottom). Mean ± SD and *P*-values of *N* = 3 in each group are shown. Two-way Student's *t*-test. subR: subretina. **c**, **d** 3-month-old KO retinal staining shows the subretinal microglia expressed DAM markers -TREM2 and galectin-3 (Gal3) (arrows in **c**), activated microglial marker CD68, and P-αSyn (arrows in **d**). **e** FIB-SEM images of 3-month-old KO retinas show microglia (blue) in the OPL (top) and subretinal region (bottom) contain abundant lipofuscins (Lf). Lipofuscin with a fingerprint pattern seen in subretinal microglia is highlighted in the inset (yellow arrow). Mitochondria and horizontal cell terminals are shaded in purple and yellow, respectively. **f** *En face* views show the Iba1-labeled subretinal microglia (adhered to the apical side of 3-month-old KO mouse RPE sheets) had bright AF and Gal3 staining. **g–i** *En face* views of the subretinal microglia show they had P-αSyn, visualized by Alexa 647 secondary antibody (false color to magenta), overlapped extensively with AF granules (visualized in green channel), in KO mouse RPE sheets, either untreated (**g**) or treated with proteinase K (**h**). **i** Pearson's coefficients show the overlaps between P-αSyn and AF were comparable between untreated (−) and proteinase K (PK) treated (+) samples. Mean ± SD. Two-way Student's *t*-test. *N* = 6 and 7 imaging fields for untreated and treated samples, respectively. **j** Representative electron micrographs of subretinal microglia of 3-month-old KO show that P-αSyn immunogolds were often found in LB-like inclusions. The enlarged view (of 2 boxed areas) highlights the P-αSyn immunogolds labeled lipofuscins (yellow arrows). Representative images of 3 and 2 independent repeats of (**a–d**, **f–i**) and (**e**, **j**), respectively, are shown. Source data of (**b**, **i**) are provided as a Source Data file.

laser ophthalmoscopy (cSLO) imaging system provided an even more sensitive system for detecting bright AF foci in KO fundi (Fig. 8b). Abundant bright AF foci appeared around 3 months of age in KO mice and continuously increased over time (Fig. 8b, c). Similar bright AF foci were not observed in the Ctrl at any age. These results suggest that the fundus AF foci is a non-invasive proxy for histologically detected αSyn lesions, which coincide with the onset and progression of retinal degeneration.

The weak and diffused AF observed in the Ctrl mouse fundi is known to be derived from the bisretionids, primarily the N-retinylidene-N-ethanolamine (A2E)[72,73]. Bistronoid metabolites are originally synthesized in photoreceptors, as a phototransduction byproduct, and in RPE where they eventually accumulate (Supplementary Fig. 7a). High-performance liquid chromatography (HPLC) studies showed that Ctrl and KO mouse eyes (3-month-old, dark-adapted) expressed comparable levels of A2E and several other bisretinoid metabolites (i.e., A2PE, A2PE-H2, and all-trans-retinaldehyde dimer conjugated with phosphatidylethanolamine) (Fig. 8d). These biochemical studies ruled out bisretionids as a major contributor to the AF in the KO mouse retinas.

The retinoid profiling studies also revealed that the 1-month-old KO (vs. Ctrl) had decreased 11-*cis*-retinaldehyde (11cRal; in dark-adapted eyes, Fig. 8e). 11cRal is the visual chromophore of rhodopsin

photopigment; its reduction agrees with the reduced rhodopsin protein and shorter OS. In contrast, the same-age KO had increased all-*trans*-retinaldehyde (atRAL; in dark and light-adapted eyes; Fig. 8f). atRAL is a toxic intermediate[74,75], its increase might contribute to the cellular death of VPS35-KO rods. The level of several trace species of the retinoids (i.e., at-retinyl palmitate,11-*cis*-retinol, and all-*trans*-retinol) expressed by KO and Ctrl were indistinguishable (Supplementary Fig. 7b, c).

## Discussion

VPS35 is an essential component of the retromer complex[76]. The whole-body knockout of Vps35 is embryonic lethal[77]. We investigated the cell-autonomous function of VPS35 in rods using a conditional KO approach because VPS35 is broadly expressed in many retinal cell types, as seen by immunostaining (this paper) and scRNAseq dataset[78–80]. We too experimentally validated the ONL-specific Cre expression and provided comprehensive immunoassays validating rod-specific loss in VPS35 in Rod$^{\Delta Vps35}$ mice. A previous mouse model with β-galactosidase under the control of Vps35 promoter had retinal staining of x-gal predominantly in ganglion cells[81]. This correlates with the scRNA dataset that ganglion cells express the highest level of Vps35 mRNA in retinas[78–80], though we surmised that such discrepancy might be also contributed by additional factors, including the gene expression altered by the x-gal reporter insertion.

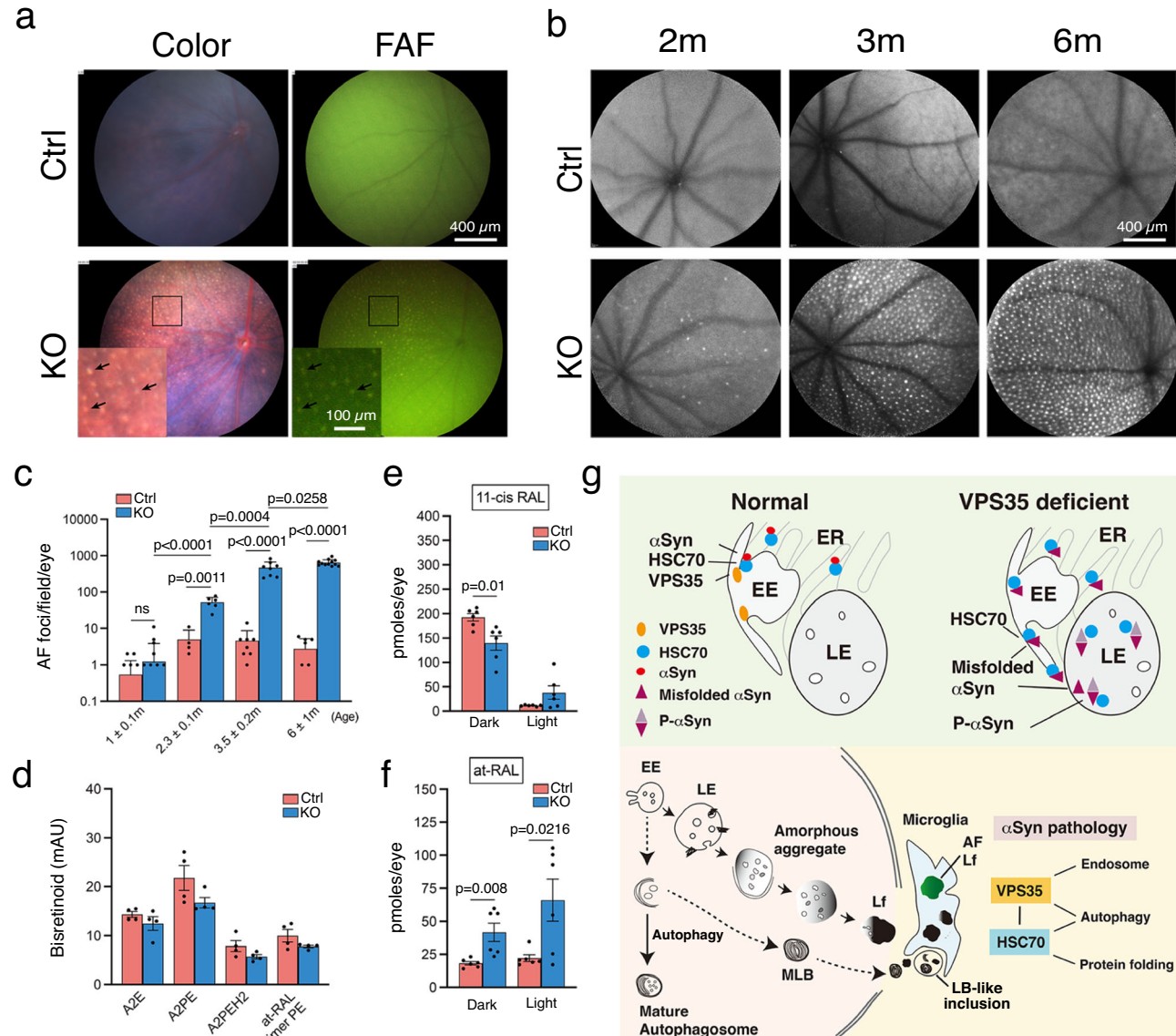

**Fig. 8 | Fundus images and retinoids profiling of Rod^ΔVps35 retinas and working model of VPS35-deficiency caused αSyn pathology. a** Color and auto-fluorescence (FAF) fundus images of 3-month-old live mice. Arrows in insets mark the matching white and AF dots. **b** Age-matched cSLO fundus images show a progressive increase of AF foci appeared in KO whereas weak and diffusive AF are detected in Ctrl of all tested ages. **c** Quantification of cSLO-detected AF foci in age-matched Ctrl and KO. Mean ± SD, and p-values are shown. $N = 13, 4, 8, 8$ eyes (Ctrl) and $N = 21, 6, 8, 12$ eyes (KO) used for $1 ± 0.1, 2.3 ± 0.1, 3.5 ± 0.2, 6 ± 1$-month-olds, respectively. Two-way Student's t-test. **d–f** Retinoid profiling. Histograms show the expression level of bisretinoid species (3-month-old, dark-adapted mouse eyes; $N = 4$ eyes) (**d**) as well as 11cRAL (**e**) and atRAL (**f**) (1-month-old, dark- and light-adapted mouse eyes; $N = 6$ eyes from 3 mice of each genotype). Mean ± SEM and P-

values are shown. Two-way Student's t-test. **g** Our data support a working model that VPS35 is the node of the three pathways (endosomes, autophagy, and protein folding) involved in αSyn pathology. VPS35 deficiency accumulates HSC70 and misfolded αSyn and, hence, P-αSyn in the LE lumens. The LE luminal aggregates that failed to be digested are also rich in lipids; they gradually form electron-dense amorphous aggregates and then lipofuscins. The lipofuscins and membranous whorls derived from the impaired macrophagic flux (i.e., MLBs) escaped from VPS35-deficient rods have the potential of forming LB-like inclusions. Microglia had P-αSyn+ lipofuscins and LB-like inclusions engulfed are activated, exhibiting visible AF and high mobility. Representative images of 3 and 2 independent repeats of (**b**) and (**a**, **d–f**), respectively, are shown. Source data of (**c–f**) are provided as a Source Data file. ER endoplasmic reticulum; Lf lipofuscins.

The longitudinal studies of Rod^ΔVps35 mice showed rod synapse loss occurred much in advance of rod cell death in these mice. The former was supported by the early morphologic change of rod terminals and the rod ERG signal decline. The decline of both rod and cone ERG is commonly observed in RP patients and many RP mouse models even though the rod is a primary lesion site[82–84]. This is because rods and cones are electrically and metabolically coupled[85]. The cones rely on rods for optimal light transduction and metabolic fitness. Therefore, the reduced cone ERG observed in Rod^ΔVps35 mice is likely to be a secondary response to sickness of the rods. Like PD patients, Rod^ΔVps35

mice also tendered contrast sensitivity decrease in behavior tests for visual acuity.

The OS dysmorphogenesis and OS protein mislocalization are also commonly observed in RP mouse models[30]. The OS disc disorganization and the disappearance of a large cohort of OS proteins in Rod^ΔVps35 mice indicate that these proteins are mislocalized and, hence, sent for degradation. This directly supports the importance of endosomes in the transport of OS proteins, a model previously proposed based on indirect evidence[86–88]. Retromer mutation in flies causes rhodopsin retention in LE/Lys and kills photoreceptors[89,90]. The fly

rhodopsin resides in the microvillar plasma membrane, while mammalian rod OS is a modified primary cilium[30]. VPS35 has recently been reported to be required for the cilium elongation[91], as well as the ciliary localization of PC1 and PC2[92,93].

Early synaptic loss is a pathological feature of PD and other synucleinopathies. Several components that are genetically linked to PD, such as αSyn (PARK1, PARK4) and the HSC70 co-chaperone auxilin (PARK19)[94], have an established role in SV recycling[16–18,21]. The SV recycling function of VPS35 has been reported in flies but not in mammals[95,96]. The LE sequestering of SVs, the declined ERG signal, and the diminished Ribeye staining in VPS35-KO rod terminals cohesively suggested that VPS35-deficient rods had impaired SV recycling. Identifying the "synaptic active zone assembly" and "SV recycling" in the pathways significantly enriched in the VPS35 interactomes (Fig. 6a, b) independently endorses VPS35's synaptic roles in mammalian rods.

Dysregulated endosomal trafficking, selective microautophagy, and protein quality control have been linked to αSyn pathology. VPS35's involvement in the first two cellular processes has been extensively characterized[46–48,53,97–102]. The enriched pathways of VPS35-IP proteomics (e.g., protein processing in ER, protein (re)folding, proteosome, supramolecular complex) implicate that the VPS35 is connected to a protein network involved in protein quality control. We focused on investigating its interaction with HSC70 and showed that VPS35-deficiency-caused LE sequestering of HSC70 and misfolded αSyn, accumulating P-αSyn aggregates. We propose that prolonged HSC70 sequestering subverts its performance in protein folding and SV recycling. A vicious cycle formed between the increasing misfolded αSyn (and other proteins) and the decreasing catabolism (due to VPS35-deficiency-caused missorting of proteins such as LE degradative enzymes[103]) exacerbates the accumulation of lipid- and P-αSyn-rich aggregates (Fig. 8g). The lipofuscins, together with MLBs and other membranes, constitute the cores of future LB-like inclusions (Fig. 8g). The presence of CD63 in LB-like inclusions is of great interest as CD63 has an active role in amyloid fibril formation in LEs[104].

Our studies comprehensively show Rod[ΔVps35] mouse retinas exhibit the salient features of αSyn pathology including the emergence of LB-like inclusion as well as the propagation and spreading of P-αSyn aggregation. Inclusions featured incredible similarity to human LBs by expressing P-αSyn, ubiquitin, lipofuscins, and membranes were detected in photoreceptor and subretinal layers of KO. Their retinas express increasing P-αSyn in all layers as well as detergent-insoluble and proteinase K-resistant P-αSyn aggregates. EM images provide visual proof of rod-to-glia transmission of lipofuscins which bind to P-αSyn. The αSyn-lipofuscin association has also frequently been found in human PD brains[20,55,105]. Taken together, we proposed, in contrast to the historic view that lipofuscin is a nonspecific AF "bystander" product of aging, lipofuscins act as a "sink" of misfolded αSyn. The buildup of lipofuscin during senescence is a risk of αSyn lesion.

In human brains, despite the ubiquitous expression of αSyn, the LBs preferably appear in certain susceptible neuronal cell types[106]. Here, despite the broad increase of P-αSyn in all retinal layers, the restricted localization of LB-like inclusions in the photoreceptor layer suggests that additional lipid/membrane-rich components derived from the atrophic neurons (rods in our case) are required for the LB genesis. This model helps to explain the concerted contribution of deranged endosomal trafficking, microautophagy, and protein quality control in causing αSyn pathology.

Emerging data is demonstrating that αSyn monomer and truncated monomer are linked with PD mutations, PD patients, and the αSyn fibrillization and LB formation in various PD models[107–112]. We too found the unusual increase of monomer and truncated monomer of αSyn as well as oligomers in the TX-100 insoluble pool of KO retinas. The TX-100-soluble αSyn we observed in both Ctrl or KO retinas migrated as tetramers and dimers. Emerging data revealed αSyn is

physiologically folded as tetramers[108,113–115] whereas αSyn dimers have been linked to oxidative stress[116]. We surmise the dimerization of retina-expressed αSyn is related to the oxidized milieu of the retina.

Some lipofuscins in the IS (Fig. 4a) and subretinal microglia (Fig. 7h) of Rod[ΔVps35] retinas had a fingerprint pattern. These fingerprint lipofuscins resemble the pathological hallmark of Batten disease, patients with Batten disease feature early RP and parkinsonism[117–119]. Batten disease is a lysosomal storage disease caused by mutations of CLN3[120,121]. CLN3 functionally interacts with VPS35 in modulating protein trafficking and autophagolysosome maturation[121,122]. Disease-associated CLN3 mutations cause increased expression of αSyn[123,124]. Therefore, we surmise the etiology underlying the αSyn pathology and retinal degeneration in Rod[ΔVps35] overlaps with that in Batten disease. Taken together, these insights give credence to synucleinopathy as a mechanism that underlies retinal degeneration in PD, RP, and Batten disease.

An interesting side observation: in the selective autophagy pathway, HSC70 binds and escorts its client proteins to the Lys for chaperon-mediated autophagy (CMA) -mediated degradation[125]. Alternatively, HSC70 undergoes endosomal microautophagy (eMI)-mediated degradation alongside its client proteins in the LE lumen[29,60,125,126]. Unlike CMA, which has been extensively characterized, the physiological relevance of the eMI remains unknown. The lack of Lys and the LE anomaly observed in the VPS35-KO rods indicate that eMI is the predominant selective autophagic degradation pathway physiologically used by rods.

Sustained microglial inflammation has been implicated in the retinal pathology in RP[127], and Alzheimer's animal models[128–131]. To date, the stimuli that reprogram the homeostatic microglia to pathological microglia in the retina remain unclear. Our investigations of Rod[ΔVps35] mice suggest the internalization of rod-spitted lipofuscin, protein aggregates, and the subsequently formed inclusions might activate the microglia, which had changing cell shape, becoming more mobile and expressing a host of inflammatory and DAM molecules.

To date, non-invasive diagnosis of PD remains a pressing need. The recently validated early PD diagnosis is based on the detection of αSyn in the spinal fluid, which is obtained through an invasive lumbar puncture procedure[132]. The current study shows that cSLO-detected bright AF foci can serve as a trackable and quantifiable surrogate for the P-αSyn pathology in the mouse retina. The possibility of retinal AF foci as a non-invasive biomarker for diagnosing synucleinopathy-associated diseases warrants investigation.

The onset of fundus AF dots in Rod[ΔVps35] retina coincides with the start of retinal degeneration, cementing the idea that bright AF dots seen in certain forms of RP patients can be a mode of tracking disease progression[133,134]. The fundus AF dots found in RP mouse models and aged mice were generally assumed to be the bisretinoid-rich OS ingested by the subretinal microglia[135–139]. This is not the case for the increased AF in Rod[ΔVps35] retinas, supported by unchanged bisretinoid levels and the absence of OS fragments in the subretinal microglia, by a thorough inspection of the FIB-SEM stacked images of the subretinal region (Supplementary Fig. 8).

The majority of PD patients exhibit at least one of several possible visual symptoms including reduced visual acuity, spatial contrast sensitivity, depth perception, and color vision, and visual hallucination[50–52]. The respective contribution of the retina versus the brain to various visual symptoms observed in PD patients remains unclear[1–9,78]. However, the retinal thinning and the detection of P-αSyn inclusions/neurites in postmortem PD patient retinas[12] suggest the presence of retinal pathology in PD. The PD mouse model TgM83 (overexpressing human αSyn A53 mutant and seeded with old brain homogenates), which only becomes clinically ill at 12–16 months of age, shows increased retinal expression of P-αSyn staining, retinal degeneration, and microglia activation as early as 5–8 months of age[140]. This, together with the current study, spotlights the retina as a valuable

central nervous system component to scrutinize the mechanism of synucleinopathy.

Previous PD models develop αSyn aggregates very late (>12-month-old) and, even then, αSyn is introduced exogenously or by transgenic overexpression of αSyn[7,141–145]. In contrast, in our Rod$^{\Delta Vps35}$ mouse retina, several αSyn lesions developed as early as 3-month-old and, more importantly, the aggregates are derived from endogenous αSyn. The early timing of the onset and progression of the αSyn pathology and neuronal dystrophies (e.g., synaptic loss, degeneration, and microglial activation) in these mice provides an ideal window of opportunity for mechanistic studies. Even better, these pathological manifestations are fully penetrant and can be longitudinally tracked using non-invasive tests like ERG, SD-OCT, and cSLO. Using a virtual optomotor system that measures reflexive optomotor responses[146], young Rod$^{\Delta Vps35}$ mice showed a trend of decreasing visual acuity. Previously, mice that had αSyn overexpressed in the inner retinas, through intraocular virus injection, exhibited significantly decreased visual acuity in a water maze task which measures perceptual visual acuity[147]. In the future, it will be interesting to test Rod$^{\Delta Vps35}$ mice using water maze tasks.

Finally, the relatively thin tissue of the retina which has nuclei and synapses organized in compacted layers provides technical advantages for stereology-based quantification. Given the accessibility of the retina, we further propose that the Rod$^{\Delta Vps35}$ mouse line is a valuable tool for testing strategies for ameliorating the αSyn lesion and its associated neuronal functional decline. The application of this ocular model in investigating brain-related PD-associated clinical histopathology and symptoms is a future interest.

## Methods

All animal experiments were approved by the Weill Cornell Medicine Institutional Animal Care and Use Committee (animal protocol 0605-490 A).

### Generation and genetic characterization of Rod$^{\Delta Vps35}$ mice

The iCre[75] mice[40] (from Dr. Ching-Kang Chen) and Vps35$^{f/f}$ mice[77] (from Dr. Wen-Cheng Xiong) were bred on the background of C57BL6/J (from Jackson Lab). We crossed iCre$^{+/-}$: Vps35$^{f/f}$ males with Vps35$^{f/f}$ females to generate age-matched iCre$^{75+/-}$; Vps35$^{f/f}$ (KO) and iCre$^{75-/-}$; Vps35$^{f/f}$ (Ctrl) littermates for experiments. We experimentally confirmed our mouse stocks have the Rpe65Met450 allele[148] and do not have the retinal degeneration 8 (rd8) allele (PCR genotype using Crb1 primers). All animals were housed in an animal facility at Weill Cornell Medicine at a relatively constant temperature of $21.5 \pm 1\,°C$ and humidity of 30%–70%, 12:12 h light: dark photoperiod; γ-irradiated standard chow and acidified reverse osmosis water provided ad libitum. The ages of animals were listed in the legends of each experiment. Our pilot studies showed that the number of cSLO-detected AF foci is indistinguishable between females and males. Therefore, an equal ratio of male and female mice was used for all the experiments. All mice in these studies were euthanized by $CO_2$ inhalation.

### Reagents

All antibodies, primers, and plasmids used in this study are listed in Supplementary Tables 5, 6, and 7, respectively.

### Fundus photography

For imaging mouse pupils were dilated with 2.5% Phenylephrine Hydrochloride Ophthalmic Solution (Akorn) and 1% Tropicamide Ophthalmic Solution (Akorn) for 5 min. The mouse was then anesthetized by isoflurane inhalation and Refresh Lubricant Eye Drops (Allergan) were put in both eyes to prevent them from drying during image acquisition. Fundus and SD-OCT scans were acquired using the Micron III-OCT2 system (Phoenix Research Labs). Animals were placed on a stereotaxic stand and the camera advanced till it coupled with the eye.

Fundus and OCT images were acquired of the left and right side of the eye keeping the optic nerve head (ONH) as a reference point. The thickness of the retinal layers was measured approximately $400\,\mu m$ away from the ONH using the software InSightv2.0.6080. cSLO fundus imaging was acquired by Spectralis HRA + OCT (Software Spectralis v6.3a; Heidelberg Engineering, Heidelberg, Germany). Focusing on the outer retina, images of 30 frames per fundus were taken once again keeping the ONH as a reference point. The AF dots were quantified manually.

### ERG analysis

The ERG measurements were done using the Espion e2 Visual Electrophysiology System (Espion V6, Diagnosys). Overnight dark-adapted mice were anesthetized with an intraperitoneal injection of a mixture of ketamine (100 mg/kg) and xylazine (20 mg/kg). The eyes of anesthetized mice were dilated with 2.5% Phenylephrine Hydrochloride Ophthalmic Solution and 1% Tropicamide Ophthalmic Solution before ERG. Body temperature was maintained at 37 °C with a heating pad. A reference electrode was inserted subcutaneously between the eyes, a ground electrode was inserted subcutaneously at the tail base and electroretinograms were recorded from both eyes using gold wire loop probes. Both eyes had contact corneal electrodes held in place by a drop of Gonak solution (Akorn). ERG was performed using flashes. Scotopic and photopic ERG were used to measure rod- and cone-mediated electrical response of the retina to light, respectively For scotopic recording, the stimulus was presented with intensities ranging from 0.001 to 3 cd·s·m$^{-2}$. For photopic recording, mice were light-adapted to 30 cd/m$^2$ background intensity for 7 min before recordings, the stimulus was presented at 1 Hz for 10 s with intensities from 0.3 to 100 cd·s·m$^{-2}$. Eight recordings were averaged per light intensity. The a-wave represent the photoreceptor activity, whereas the b-wave reflects of the amplified signals mediated by the bipolar and Müller glia cells.

### Visual acuity assays

Visual acuity was determined by optomotor reflex measurement in the OptoDrum (Striatech) virtual optomotor system. The system consists of a small chamber ($54 \times 54 \times 30$ cm) with four screens (23.8", $1920 \times 1080$ pixel resolution, in-plane switching [IPS] screen) surrounding a platform, and the bottom and the top of the chamber is covered with mirrors. Head movements of unrestrained mice sitting on the elevated are tracked from above by an infrared-sensitive digital camera while a rotating pattern of black and white vertical bars was displayed at different spatial frequencies controlled by the software OptoDrum v1.5.0. For measurement of scotopic visual acuity, the animals were dark-adapted overnight. Four neutral-density filter foils of 1.2 log units per filter were placed in front of each of the 4 screens and any light source in the room was covered with red transparent Plexiglas to ensure darkness. The velocity of the moving bars was set to 12°/s, and the contrast was set to 100%. The recorded head movements were analyzed by the Optodrum software, which evaluated the mice for their stimulus pattern, and rated the trial as fail or pass. Animals were measured on three different days and an average of the spatial frequency (cycles/degree) of each animal on three different days was used for assessment.

### Tissue processing and immunohistochemistry

For histological staining of the retinal sections, marked eyes were placed in Excalibur's Alcoholic Z-Fix (Excalibur Pathology) at room temperature for >48 h and then processed for paraffin embedding, sectioning (4-μm-thick), and staining with Hematoxylin and eosin dye. The H&E sections were cut along the inferior and superior direction of the eyeball. The thickness of each layer was measured every 300 μm from the optic nerve head of the retinal panorama. The measurements of the retinal layer thickness were calculated

from the cross-sectional images using the ruler tool of Adobe Photoshop 25.9.

For immunostaining, eyecups were submerged fixed in 4% paraformaldehyde (PFA) for 1–4 h at room temperature. The eyecups were embedded in agarose II™ (Fisher Scientifics) and vibratome sectioned (40-µm-thick). To prepare flat mounts, eyecups were fixed with 4% PFA for 1–4 h, the anterior segments removed and the retina separated from RPE-choroid-sclera cups/sheets. The free-floating technique was used for antibody staining. In some experiments, the tissues were treated with 10 µg/ml of proteinase K (SIGMA) at 37 °C, 10 min before staining, as described[71]. Confocal images were taken by a Zeiss LSM880 or Zeiss spinning disc Observer Z1 confocal microscope. For presentation purposes, images were processed using Adobe Photoshop 25.9 & Adobe Illustrator 28.5. Lambda scans (Leica Stellaris 8) were used to determine the peak of fluorescence of AF.

For quantification, confocal images taken from Ctrl and KO mice using matched acquisition parameters were processed and compared in parallel ($N = 3$, 3 fields of similar retinal regions per mouse). (1) Line scan was used to quantify the VPS35 signals across the different retinal regions using ImageJ2 (V2.14.0/1.54f (plugins)). A line traversing the retina was drawn using the freehand line tool, saved using the Region of Interest (ROI) Manager tool (Analyze > Tools > ROI Manager), and analyzed using Analyze > plot profile. The values for the different regions were segregated and the average intensity of the positive signal was calculated for presentation. (2) The number and area of EEA1, Lamp1, CD63, LC3, Ribeye, αSyn, and P-αSyn were quantified using ImageJ. The threshold for the ROI was adjusted to cover positive areas and quantified by choosing Analyze > analyze particles > display results. (3) The relative synaptic contact density was calculated by the number of mGluR6-labeled synapses (across 100 µm-width of OPL in stained retinal slices) and normalized by the number of nuclei in the ONL (ImageJ). (4) We used the ImageJ JACoP plugin to calculate Pearson's correlation coefficient of the overlapped signal of LC3 and Lamp1 and Pa-Syn and AF. (5) We used the described method[149] to measure the expression of IL-1β and CD68 in microglia. Briefly, the area of 4 or more randomly chosen Iba1-labeled microglia from one confocal plane was measured using Fractal Analysis. The staining intensity of IL-1**β** and CD68 in each Iba1+ microglia was measured from the same confocal plane. The threshold was adjusted to cover the stained area of each cell and by choosing the Analysis > Measure tool, the Raw Integrated Density was determined. The relative intensity was obtained by dividing the Raw Integrated Density by cell area.

## Transmission EM, FIB-SEM, and ImmunoEM

Mice were transcardially perfused with 10 ml of heparin saline (20 units/ml), 20 ml of 4% PFA/3.75% acrolein (Polysciences) in 0.1 M phosphate buffer (pH 7.4) and 60 ml of 4% PFA in 0. 1 M phosphate buffer. Eyecups were prepared and stored in 2.5% glutaraldehyde, 4% PFA /0.1 M cacodylate buffer at 4 °C. 120 µm-thick vibratome sections obtained from small pieces of eyecup (~2 × ~2 mm) embedded in 5% agarose II™/0.1 M cacodylic acid were processed for en bloc fixation and osmium tetroxide-thiocarbohydrazide-osmium (OTO) staining method (to enhance the contrast of lipid components), as described[150]. For the OTO method, after several washes in ice-cold 0.15 M cacodylate buffer containing 2 mM calcium chloride, specimens were incubated with 1.5% potassium ferrocyanide, 2 mM calcium chloride, and 2% osmium tetroxide in 0.15 M cacodylate buffer, pH 7.4 for 1 h on ice. The sections were then treated with 10 mg/ml of thiocarbohydrazide (Polysciences) solution for 20 min followed by 2% osmium tetroxide fixation for 30 min, both at room temperature. The en bloc-stained tissues were dehydrated with graded ethanol and embedded in Epon. Ultrathin sections (72 nm) were collected on G400-Cu grids (Electron Microscopy Sciences) and were examined under a TECNAI microscope for conventional transmission EM analysis. For FIB-SEM, we subjected the en bloc-stained retinal tissue Epon blocks to be precision milled

every 10 nm (z-resolution) and with a pixel resolution of 2.5 and 5 nm/ pixel in x and y, respectively. The block face images were collected by the FEI Helios NanoLab 650 microscope. Images were processed using Serial Sections Alignment Programs of IMOD/eTomo to correct drifting caused by the 30° angle from the block face during imaging.

The distribution of lipofuscin, amorphous, and MLB were manually scored (by two individuals with trained eyes) through 3-10-µm image stacks. The volume of the stack was quantified and presented in counts /mm³. For immunoEM, retinal vibratome sections (110 µm-thick) were processed by a "freeze-thaw" protocol. i.e., the sections were incubated in 25% sucrose/3% glycerol in PBS for 30 min at room temperature before transferring to −80 °C for 10 min and then thawed at 37 °C. The sections were quenched (50 mM NH₄Cl in PBS for 10 min), permeabilized (0.05% Triton X-100 in PBS for 10 min), blocked (0.5% BSA in PBS for 30 min), incubated with primary antibody (0.5% BSA in blocking buffer) for 2 overnights at 4 °C, followed with biotinylated donkey anti-rabbit (Ab in 0.5% BSA in blocking buffer) for 1 overnight at 4 °C. The sections were then washed 0.8% BSA/0.2% (w/v) cold water fish gelatin (Amersham Biosciences RPN41) supplemented PBS for 10 min followed by incubation with streptavidin UltraSmall Immunogold (Electron Microscopy Sciences, cat#25260; 1:50) in 0.8% BSA/ 0.2% (w/v) cold fish gelatin-supplemented PBS at room temperature for 2 h. To improve the visibility of labeled ultrasmall gold particles (<0.8 nm), we used sliver enhancement to intensify and enlarge the gold particles (by metallic silver deposition) to 3−20 nm. The heterogeneous amount of silver deposits accounts for the variable sizes of silver-gold particles. The stained sections were post-fixed with 2% glutaraldehyde for 10 min followed by silver-enhancement using Aurion R-Gent SE-LM (Aurion Inc.) for 17 min, followed by the en bloc fixation and staining as described above.

## Immunoblotting assays and quantification

For OS protein detection, mouse retinas removed from the remaining eyecups were lysed in RIPA buffer (50 mM Tris-HCl, pH 7.4, 150 mM NaCl, 0.5% NP40, 0.5% Na-deoxycholate, 1 mM EGTA, 5% glycerol, 0.5% CHAPS) plus protease and phosphatase inhibitors cocktails. Protein concentration was measured by the BCA method (BioRad). Postnuclear supernatants (after $15{,}700 \times g$, spinning 15 min at 4 °C) were heated up 70°, 10 min in SDS-sample buffer and electrophoresed on NuPAGE 4%–12% protein gel (Thermo Fisher) and transferred using iBlot™ (Thermo Fisher).

For αSyn detection, two mouse retinas were placed in 400 µl PBS (plus protease and phosphatase inhibitors cocktails), passing through the 261/2-gauge needle, sonicated (30% input, 60 cycles) on ice/water bath, and centrifuged ($500 \times g$, 15 min, 4 °C). The supernatants were incubated with protein-G dyna beads (Thermo Fisher) conjugated with B6-30 antibody (that recognized the extracellular domain of rhodopsin) to remove OS fragments. Subsequently, the retinal PBS sups were incubated with an equal volume of 2% Triton X-100 supplemented STE buffer (150 mM NaCl, 50 mM Tris, pH 7.5) plus protease and phosphatase inhibitors cocktails followed by sonication (30% input, 20cycles) and centrifugation ($15{,}700 \times g$, 30 min, 4 °C). The supernatants are the TX-100 soluble fractions. The pellets were washed once with room temperature 1%TX-100 STET buffer and once with STE. The pellets were dissolved in 100 µl of 2%SDS/8 M urea at room temperature for 10 min followed by sonication (30% inputs, 20 pluses). TX-100 insoluble/Urea soluble fractions. The protein concentration of the TX-100 soluble retinal lysates was determined by the BCA method. Because 2%SDS/8 M urea is incompatible with the BCA method, the protein concentration of the TX-100 insoluble retinal lysates was determined by a serial diluted samples dotted on blots (alongside with known concentration of protein standard) followed by Total Revert™ 700 Total Protein Stain and Odyssey scanning to determine the total protein concentration. 0.1–1 µg of TX-100 soluble and insoluble samples were denatured (by heating 37 °C for 20 min,

70 °C 10 min, boil for 5 min) in sample buffers before electrophoresis on NuPAGE 4%–12% protein gels, transferred (BioRad Turbo program, 25 mV, 1 Amp 10 min). Standard immunoblotting assays using Infrared-dye secondary antibody followed by Odyssey scanning/quantification. Uncropped full blots of all figures were shown in Supplementary Figs. 9 and 10.

## RNA extraction and qRT-PCR

For real-time qPCR, we isolated mouse retina RNA (using Qiagen RNeasy mini kit) and generated first-strand cDNAs by using Oligo(dT) 20 primers and SuperScript™ III First-Strand Synthesis System (Thermo Fisher Scientific). Triplicates of 20 µl PCR reactions (Power SYBR™ Green PCR master mix) (Applied Biosystems) were run on the StepOnePlus Real-time PCR system (Applied Biosystems). The cycle threshold (Ct) values of interesting genes were normalized to the housekeeping gene Gapdh. The genes' relative expression levels were calculated by the 2(−ΔΔCt) method.

## Immunoprecipitation, LC–MS proteomics, and enrichment pathway analyses

The mouse retinas were harvested and lysed by RIPA (50 mM Tris-HCl, pH 7.4, 150 mM NaCl, 0.5% NP40, 0.5% Na-deoxycholate, 1 mM EGTA, 5% glycerol) supplemented with 0.5% CHAPS plus 1 mM PMSF and cocktails of proteinase and phosphatase inhibitors. After being sonicated in an ice/water bath, the cell lysate was centrifuged at $15,700 \times g$ (15 min, 4 °C) twice. The supernatant was incubated with Dyna beads for 1 h. Afterward, the supernatant was incubated with control goat IgG or VPS35 antibody for 1 h in RT, followed by incubating with Dyna beads for 2 h in 4 °C. After washing, the beads were eluted with Elution buffer (50 mM Tris, 150 mM NaCl) containing VPS35 c-terminal peptide (c-SPESEGPIYEGLIL, Everest Biotech). The elution samples were subjected to analysis by Western Blot or LC–MS.

The samples were loaded to the 1.5 mm 10-well 4–12% Bis-Tris precast Gel (Thermo Fisher). After running the samples into the gel for 1 cm, staining the gel with Coomassie blue for 1 h. Wash the gel with Methanol/ Acetic overnight. The in-gel samples were excised and transferred into a clean Eppendorf tube. The in-gel trypsin digestion, desalting, and LC–MS/MS were performed by WCM Proteomics and Metabolomics Core Facility. For protein identification and quantification, the raw data were processed by MaxQuant v2.6.1. The Uniprot mouse protein database was searched for comparing the MS data (see Supplementary Table 1). 196 human homologs were mapped from 211 of total mouse molecules from three experiments. We conducted the pathway enrichment analyses based on Fisher's test, GSEA, and DAVID.

## Recombinant protein and pull-down assay

E. coli bacterial lysates containing His-HSC70 or GST-VPS35 full-length, GST-VPS35-N-terminal, and GST-VPS35-C-terminal fragments were homogenized in lysozyme (2 mg/ml), protease inhibitors supplemented PBST (PBS + 1% TX-100), sonicated (30% input, 2 min, on ice/water bath), and centrifuged ($15,000 \times g$, 15 min, 4 °C). The supernatants were analyzed on Pageblue stained SDS–PAGE gels, and the yield of the recombinant proteins was quantified by Odyssey Infrared Imager. The PBST bacterial lysates containing roughly equal amounts of the GST-VPS35 fragments were incubated with glutathione sepharose 4B beads for 1 h. After 3 PBS washes, His-HS70 containing PBST lysates were added to the GST proteins-conjugated beads for 1 h at 4 °C. Afterwards, beads were washed with PBST and PBS, and eluted with 100 mM reduced glutathione. The elutes were subjected to electrophoreses and immunoblotting assays as described above.

## Cell studies: transfection, retinal dissociation, staining and imaging

661W cells are a photoreceptor-like line commonly used to study RP[151–157]. 661W cells (a gift from Muayyad Al-Ubaidi; RRID:CVCL_6240)

were transfected using the Amaxa Nucleofector system for several days (indicated in legends) and followed by immunostaining or immunoblotting assays. Briefly, 10 µg plasmids were introduced into 1–2 million cells. In some experiments, myc-αSyn, shRNA: Flag Rab5(Q79L) were mixed in a ratio of 2:2:1. Rab5(Q79L) was co-transfected, as described[49,158], in order to expand the size of the LE to increase the visibility of its lumen contents. Doxycycline (0.4 µg /ml) was added to the cultures if the plasmids were built on a Tet-on inducible vector. Cells were permeabilized and stained in blocking buffer-PBS-C/M (PSB with 2 mM calcium chloride and 0.2 mM magnesium chloride) plus 0.5% BSA, 0.2 mg ml$^{-1}$ Na-azide, 0.3 µM DAPI, and 0.25% Triton X-100. Immunostained cells were acquired using a Zeiss spinning disc Observer Z1 confocal microscope. For the quantification of the LE-resided aggregates, we employed images acquired by a 63× oil immersion lens from Ctrl and KO. Three independent experiments were performed for each condition. For quantification, more than 100 cells were counted from >15 random fields. Granule (defined by the vacuoles) had a diameter 3-fold larger than that of Lamp1-stained vacuoles in non-transfected cells.

Mouse retina dissociation was performed as described[159] with modifications. Briefly, retinas were collected in sterile $O_2$: $CO_2$ equilibrated Earle's balanced salt solution (EBSS) and digested with papain (3 U/ml) and DNase-I (100 U/ml) (Worthington Biochemical) in EBSS for 40 min at 37 °C with gentle shaking every 10 min. Following inactivation by an equal amount of DMEM/F12 medium supplemented with 10% FBS, the cells were gently triturated with a 5 ml pipette. After letting the larger pieces settle for 2 min, the supernatant was collected and spun at $100 \times g$ for 5 min. The cell pellets were resuspended in DMEM/F12 and plated on coverslips precoated with poly-L-lysine (100 µg/ml, Sigma) and then Phaseolus vulgaris erythroagglutinin (0.1 mg/ml, Vector Labs).

## Stereology analyses of microglial morphometric parameters

Fractal Analysis for determining the average cell radius and perimeter was done using the FacLac plugin in ImageJ (Young and Morrion, 2018). Briefly, after converting 8 bit, Iba1 fluorescent stained images to greyscale, the brightness/contrast was adjusted to ensure all microglial processes were visible. Images were subjected to the Unsharp Mask with a pixel radius of 3 and mask weight of 0.6 and to the Despeckle filter to sharpen contrast and remove background noise respectively. The image was converted to binary by adjusting the Threshold. After applying the Despeckle filter again, the Close function was applied to join missing pixels between points separated up to 2 pixels, and the Remove Outliers function to delete any outlier pixels deviating above the threshold. Using the rectangular tool, an ROI was drawn to cover the largest cell and 4 or more randomly chosen microglia were cropped out from a section image in a new window. An outline of the astrocyte was obtained by choosing the Process > Binary > Outline function. The outlined images, after deleting any background non-specific pixels not associated with the cell, were then subjected to Fractal Analysis using Plugins > Fractal Analysis > FracLac function. The size of the rectangular ROI for choosing microglia was kept constant throughout the analysis.

## HPLC assays of retinoids and bisretinoids

One- and 3-month-old mice were dark-adapted overnight before sacrifice. The eyes were enucleated, snapped frozen in liquid nitrogen, and stored at −80 °C until further processing. All steps were done in the darkroom under a dim red light (Kodak Wratten 1A). To protect the retinaldehydes, eyes were homogenized in 1× PBS containing 200 mM hydroxylamine, and extracted in hexane. The samples were analyzed using previously optimized HPLC methods[160]. The identity of eluted peaks was established by comparison with spectra and elution times of known authentic retinoid standards. Retinoid amounts were quantitated by comparing their respective peak areas to calibration curves

established with retinoid standards: all-trans-retinyl palmitate (at-RP11-cis-) and all-trans-retinol (11cROL and atROL), and 11-cis- and all-trans-retinaldehyde (11cRAL and atRAL) syn- and anti-oximes. Retinoids were expressed as pmoles per eye. Bisretinoids were extracted by chloroform and analyzed as described before[161]. Specifically, pooled eyecups were homogenized in 1× PBS, washed with chloroform/methanol (2:1, v/v), and extracted with chloroform (4:3, v/v). The organic phase was isolated by centrifugation at $1000 \times g$ for 10 min, dried down under argon, and resuspended in 100 µl of isopropanol. To determine the molar amount of A2E, absorbance units corresponding to the A2E peak at 435 nm were converted to pmoles using a calibration curve with authentic standards and published molar extinction coefficient[162]. All bisretinoids (A2E, A2PE, A2PE-H2, and atRAL dimer PE) are expressed as milli absorbance units (mAU). Data are presented as means with a standard deviation of 4–6 biological samples for each genotype. Biological samples consist of a single eye ($N = 6$, three mice per genotype) and three pooled eyes ($N = 4$, six mice per genotype) for retinoids and bisretinoids extraction respectively.

## Statistics and Reproducibility

The number of biological replicates and independent repeats used are described in figure legends. Statistical differences were determined using GraphPad Prism software V10.2.2 (341), or Microsoft Excel for Mac Version 16.16.27 (201012). The statistical tests used were described in figure legends. Differences between values were considered significant when *$p < 0.05$, **$p < 0.01$, ***$p < 0.001$, ****$p < 0.0001$.

## Reporting summary

Further information on research design is available in the Nature Portfolio Reporting Summary linked to this article.

## Data availability

The authors declare that data supporting the findings of this study are available within the main text and supplementary materials. Source data main Figs. 1d, e, 2a–d, 3c, e–g, 5b, f, 6h, 7b, i, and 8c–f, Supplementary Figs. 1d, f, h,2a, c, 3c, 4b, 6c, d, 7b, c, Pearson's coefficients (related to Figs. 3h and 7h) and Supplementary Table 1 are provided with this paper. The proteomics data generated in this study have been deposited in the ProteomicXchange Pride database. Accession: PXD044890. All data are available from the corresponding author (chsung@med.cornell.edu) upon request. Source data are provided with this paper.

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

## Acknowledgements

We thank Ashleigh Raczkowski for performing FIB-SEM imaging and Dr. Sushmita Mukherjee for advice on Lambda scans. Grants from NIH RO1 EY 032966, EY 029428, and RPB support this work (C.-H.S.). C.-H.S. is a recipient of the Stein Innovation Award from Research To Prevent Blindness (RPB), and an Alcon Research Award. R.A.R. is supported by NIH RO1 EY025002. M.P.L. and H.H.Y. are supported by the Intramural Research Program of the NIH and the National Cancer Institute. W.-C.X. is supported by the NIH NIA (RF1 AG045781). T.S.W. is supported by the Department of Pathology and Cell Biology at Columbia University. Proteomics was performed at Proteomics and Metabolics Core (Weil Cornell Medicine). FIB-SEM was performed at the Simons Electron Microscopy Center and National Resource for Automated Molecular Microscopy, located at the New York Structural Biology Center, supported by grants from the Simons Foundation (SF349247), NYSTAR, and the NIH National Institute of General Medical Sciences (GM103310) with additional support from NIH (RR029300).

## Author contributions

Conceptualization: C.F., J.-Z.C., N.R., R.A.R., C.-H.S. Methodology, formal analysis, and investigation: C.F., N.Y., J.-Z.C., N.N., S.I., N.R., Z.W., Z.J., W.O., H.H.Y., M.P.L., R.A.R., T.W., W.-C.X., C.-H.S.; Writing, reviewing and editing: C.F., N.Y. J.-Z.C., S.I., N.R., Z.W., W.O., R.A.R., T.S.W., H.H.Y., M.P.L., W.-C.X. C.-H.S.; Funding acquisition: R.A.R., T.S. W., M.P.L., W.-C.X., C.-H.S.; Resources and supervision: R.A.R., M.P.L., T.S.W., W.-C.X., C.-H.S.

## Competing interests

The authors declare no competing interests
