## [Peer Review File · Nature Communications]

Mutant mice with rod-specific VPS35 deletion exhibit retinal α -synuclein pathology-associated degenerationReviewers' Comments:

Reviewer #1 (Remarks to the Author):

In this study, the authors sought to determine the effect of rod-specific deletion of Vps35. VPS35 is a core component of the retromer complex, which promotes the endosome-to-Golgi recycling of transmembrane protein. There is data supporting its role in regulating excitatory synaptic transmission. VPS35 deficiency and retromer dysfunction are also implicated in the pathophysiology of Parkinson disease (PD) and Alzheimer's disease (AD). Early visual dysfunctions are commonly found in these patients. In this manuscript, the authors report that rod-photoreceptors-specific VPS35 deficiency leads to early synaptic loss followed by degeneration of photoreceptors. VPS35-deficient mice showed altered endolysosomal anatomy and autophagic function, lipofuscin deposition, and developed synucleinopathy. The authors also describe the activation of and the presence of phosphorylated α -Synuclein (P- α -Syn) in microglia. Overall, this is a well-designed study and well-written manuscript. However, there are major concerns that put into question the novelty and the premise of this study. Additional minor concerns were identified, all of which require a major and careful revision before the manuscript can be reconsidered.

Major points:

1) It is surprising that the authors choose to deplete VPS35 in rod photoreceptors. Indeed, in a previous study from one of the lead co-authors, it was shown in rather convincing ways that VPS35 is only expressed in a subpopulation of retinal ganglion cells (using mice with β -galactosidase under the control of VPS35 promoter). *This study was omitted or missing from this manuscript.

* Wei Liu, et al, Wen-Cheng Xiong (2014) Vps35 haploinsufficiency results in degenerative-like deficit in mouse retinal ganglion neurons and impairment of optic nerve injury-induced gliosis. *Molecular Brain*, PMID: 24512632.

Hence, the authors of the present manuscript should address this discrepancy and validate that their results unequivocally depend on VPS35 deletion in rod photoreceptors.

2) The depletion of rod-specific VPS35 is only shown in 3-week-old mice and only in the inner segment (IS) of the photoreceptors. To help validate their model, the authors should validate the implications of VPS35 depletion at all time points (1, 3, 6 and 9 months).

3) The depletion is only shown in the inner segment of the photoreceptors. However, there is still substantial VPS35 immunoreactivity in the ONL and OPL, likely from the photoreceptors. It would be helpful to quantify VPS35 in these layers to fully evaluate this model.

4) Rod and cone functions were evaluated by scotopic and photopic ERGs, respectively (Figure 2D and E). Surprisingly, photopic ERG response decreased in the VPS35-depleted mice. However, this model should not affect cone photoreceptors. The authors should discuss this result as it could signal some non-specific effects.

5) The microglial activation in response to VPS35 depletion is an interesting finding of the study as it shows the normal microglial response (no depletion of VPS35 in these cells) to lipofuscin or P- α -Syn, and yet, not much was discussed by the authors. Authors ought to refer to retinal microglia activation in the context of AD and retinal degeneration, which has been well characterized.

Recommended citations:

- Ni Jin, et al (2021) *Front Cell Dev Biol*, PMID: 34966736.
- Shi H, et al. (2022) *Acta Neuropathol Commun*. PMID: 36076283.
- Xu QA, et al (2022). *Acta Neuropathol Commun*, PMID: 36199154.
- Koronyo Y, et al. (2023) *Acta Neuropathol*, PMID: 36773106.

6) The authors should also discuss their findings regarding the location (retinal layer) of P- α -Syn in the post-mortem retina of patients with PD and the relevance of their model, characterized by Lewy bodies-like inclusions in and degeneration of photoreceptors.

7) The immunoprecipitation method and interaction data are sufficiently described. The authors should detail what kind of interactions are expected. Direct interactions with VPS35? Complexes containing VPS35? Are interactions detected on the entire VPS35 protein? N-terminal? C-terminal? Some of the interactors do not seem to be specific. For example, numerous histones. The authors should mention these results.

8) Even though the authors focused on HSC70 (HSPA8), it would be interesting to perform

enrichment analysis of the interactors detected for VPS35. It could give additional support of the results regarding pathways that involved VPS35.

Minor points

- 1) The legend of Figure 2D and E is difficult to understand. The authors should refer to the bar graphs or the traces and better explain a- and b-waves.
- 2) In the legend of Figure 7 (page 34, line 1084), macrophagic should be autophagic.
- 3) In the legend of Figure S2 (page 37, line 1101), (C, D) should be (B, C).
- 4) The supplementary table 1 is incomplete (page 40). Please add the remaining interactor proteins after HSPA5.
- 5) On page 40, line 1123 and on page 41, line 1124, please correct as Supplementary Table 1 and Supplementary Table 2, respectively.

Reviewer #2 (Remarks to the Author):

Reviewer #3 (Remarks to the Author):

The results are interesting and relevant not only to Parkinson's disease. The relevance of the discovered VPS35-HSC70-pasyn interaction are very interesting; its impact could be further explored by testing it in different pathological contexts, such as alpha-synuclein overexpressing models.

The main problems are related to methodological issues, which make it difficult to evaluate the robustness of the presented data and the lack of in vivo evidence of visual acuity impairment in these animals. In particular:

- Single-point histograms should be reported.
- Figure 1E and 1F: Is there any evidence in the literature, with mice of the same genetic background as this study, that the thickness of the ONL and OPL does not change in the 6- and 9-month-old control animals? Note that the scotopic ERG, in control animals, decreases already at 6 months compared with younger animals (Fig. 2D). An age-matched control should be used in all experiments.
- Immunofluorescence analyses are often not quantified by standard stereological methods; it is necessary to have them.
- Co-localization experiments (Fig. 3E) must be quantified, using unbiased counting methods (stereology).
- Figure 3D is missing images of the immunoblots. All original blots should be provided; I could not find them in the source data.
- Analysis of p62 or other substrates of autophagy should be provided to verify impairment of autophagy. In general autophagosome/lysosome fusion needs to be evaluated using appropriate assays (PMID: 33634751)
- The data on oligomeric alpha-syn are not convincing and are not quantified. Similarly, the IF for p-asyn has not been quantified. To confirm the presence of aggregates, proteinase-K resistance should be performed and evaluated.
- Control images for 5E and 5H are not reported.
- Although the ERG provides an indication that KO mice might have a visual acuity defect, specific

behavioral tasks (using an appropriate number of animals per group based on power analysis) are needed at 1 and 3 months to assess functional changes in the observed defects.

Reviewer #4 (Remarks to the Author):

The manuscript entitled " Mutant mice with rod-specific VPS35 deletion exhibit visual deficit, synaptopathy, and retinal synucleinopathy" written by Fu et al investigated the role of rod-specific VPS35 in retinal synaptopathy, synucleinopathy and visual function. They found that rod-specific VPS35 deletion causes dysregulated OS genesis, endolysosomal homeostasis, protein degradation and autophagic flux in rods as well as early rod terminal loss before rod cell death using both light microscopic and electron microscopic analyses. Moreover, proteomic analysis demonstrated a crosstalk between VPS35 and chaperon HSC70. The findings are interesting. However, the major issue in the manuscript is that some of conclusions made by the authors cannot be backed up by the current data but by speculations.

1. 661W cell line is cone photoreceptor cells. However, the authors used the cell line for rod cells. Even though VPS35 signals appeared a fraction of 661W cells (Fig. 5E), the authors still used the cell line to study VPS35-HSC70 interactions and claimed to uncover a novel crosstalk between VPS35 and HSC70.

2. For proteomics, the authors didn't even perform KEGG pathway analysis and protein-protein interaction analysis and claimed to identify HSC70 as a VPS35-interacting protein.

3. The authors generated a mutant mouse line with rod-specific VPS35 deletion and studied the role of rod-specific VPS35 in retinal synaptopathy, synucleinopathy and visual function. Yet, the authors claimed that fundus AF dots in the mutant mouse can be used as a potential early PD biomarker.

4. The authors stated that VPS35 is a hub that functionally couples the three major pathways endolysosomal trafficking, protein quality control, and selective microautophagy. The three major pathways were identified sole by imaging analyses. There is no any functional study to confirm the relationship between VPS35 and the three major pathways.

Reviewer #5 (Remarks to the Author):

In the current study, Fu et al. generated a rod-specific VPS35 knockout mouse model and addressed synucleinopathy in the eye of the animal, exhibited by microglia activation with p-a-syn and autofluorescence accumulation, which the authors suggested as a potential biomarker for PD. In addition, the authors suggested presence of VPS35-HSC70 cross talk via late endosome sequestering and a-syn aggregation. This study takes a novel approach and is very interesting in both study design and considering eye degeneration has not been studied much in the field of synucleinopathies. However, there are some important concerns that currently diminish the study.

Major critiques:

Due to the fact that VPS35 is strongly associated with autophagy, the dysregulation of autophagy, cellular toxicity, and accumulation of a-syn aggregates are expected. The manuscript appears to this reviewer to be the sole discoverer of the phenomenon. Since the authors claimed this model may apply to synucleinopathies, the authors should include data which can show the relation between this model and synucleinopathies. For example, can the authors detect a-synuclein neuropathology in the brains of aged KO mice (at least in the visual cortex)? Does the synucleinopathy animal model also show retinal degeneration? Or autofluorescence in the microglia? Is the degeneration inhibited by an a-syn knockout?

The authors claimed that there is microglia activation through infiltration into the retinae. However, microglial activation should be determined in multiple ways, such as with inflammatory cytokine gene expressions and morphological differences.

Where was a-syn was expressed in the retina? Microglia don't express a-synuclein, so internalized and accumulated p-a-syn should be exogenously internalized into the microglia. However, significant a-syn pathology was not observed in the retina. As such, the source of microglial p-a-syn should be addressed. VPS35 is also associated with a-syn degradation. However, if the rods express a-syn, why did the VPS35 knockout not induce a-syn accumulation in the rod?

Other critiques:

Introduction

Line 52 – The optical symptoms of PD is not a very well-known topic in the field. Therefore, more detailed information to introduce the topic should be included, such as the % of occurrence and visual disturbance symptoms (color, contrast defect, etc.).

Line 56 – P129, not P219 and please use same abbreviation for a-syn (Ex, a-Syn or P-aSyn).

Line 62 – Pathogenic interaction between HSC70 and a-syn has been reported, which should be mentioned.

Results

Fig 1a,b – Provide magnification and size of picture. Provide rationale for why Ribeye, snap25, and GluR6 were analyzed in the mice.

Fig 1c – Although immunostaining provided visual quantification data, a more solid and conservative analysis such as an immunoblot is required to validate the knockout of VPS35.

Fig 1d – Provide age-matched control HE staining in parallel with KO mice, which will address whether degeneration is age-related and/or progressive.

Fig 2c – According to Fig 1, control and tg did not show much difference in at one month. To this reviewer, data analysis for mice that are 3 months old or older would more appropriate. In addition, this reviewer also could indicate short and dark tube-like shape in the WT, which needs to be addressed.

Line 131 – Again, many scientists in the field are not familiar with the electroretinogram, so please provide a brief explanation of the method.

Fig S1a – Provide age-matched analysis data

Fig 3a,b – Why was OS analyzed in 1 month old mice and retinal staining done for 3 month old mice?

Fig 3d – Provide original image of western blot

Fig 3e – Data looks clear, but again control data should be provided.

Fig 3g – Western blot should be analyzed in the same blot to avoid unnecessary misunderstanding.

Fig 3h – Quantification of western blot is difficult to understand. This should be transformed as it is in Fig 3d and include individual and control data.

Lines 164-5 - Are these proteins targeted by the proteosome or autophagy? It would be beneficial to the paper if this can be validated in an in vitro model system.

Fig. 4 – Again, the FIB-SIM data of the age-matched control should be included. Appropriate quantification should also be added; for example, the numbers of SV or MLB.

Fig. 5 – Why were 3-month-old animals used for immunofluorescence staining and 6-month-old animals for immunoblot analysis? B and C seems to be KO, but were not labeled and need controls. Where does p-syn accumulate? In which cells? And how about a-syn expression?

Fig. 6 – Immunolabeling data seems reliable, but flat mount does not. To this reviewer, all signals might originate from AF structure, so would suggest double labeling Iba1 with p-syn instead of the flat mount.

Fig 7b – Please include age-matched control data.

Discussion

What is the benefits and limitations of current model compared to well-known synucleinopathy model?

The relation and application of the authors findings with synucleinopathy should be addressed.

Point-by-point responses to Reviewers' comments:

Abbreviation keys: PD: Parkinson's; P- α Syn; phosphorylated α -Synuclein; OS: outer segment; IS: Inner segment; KO: knockout; ONL: outer nuclear layer; OPL: outer plexiform layer.

Reviewer #1 (Remarks to the Author):

In this study, the authors sought to determine the effect of rod-specific deletion of Vps35. VPS35 is a core component of the retromer complex, which promotes the endosome-to-Golgi recycling of transmembrane protein. There is data supporting its role in regulating excitatory synaptic transmission. VPS35 deficiency and retromer dysfunction are also implicated in the pathophysiology of Parkinson disease (PD) and Alzheimer's disease (AD). Early visual dysfunctions are commonly found in these patients. In this manuscript, the authors report that rod-photoreceptors-specific VPS35 deficiency leads to early synaptic loss followed by degeneration of photoreceptors. VPS35-deficient mice showed altered endolysosomal anatomy and autophagic function, lipofuscin deposition, and developed synucleinopathy. The authors also describe the activation of and the presence of phosphorylated α -Synuclein (P- α -Syn) in microglia. Overall, this is a well-designed study and well-written manuscript. However, there are major concerns that put into question the novelty and the premise of this study. Additional minor concerns were identified, all of which require a major and careful revision before the manuscript can be reconsidered.

Major points:

1) It is surprising that the authors choose to deplete VPS35 in rod photoreceptors. Indeed, in a previous study from one of the lead co-authors, it was shown in rather convincing ways that VPS35 is only expressed in a subpopulation of retinal ganglion cells (using mice with β -galactosidase under the control of VPS35 promoter). *This study was omitted or missing from this manuscript.

* Wei Liu, et al, Wen-Cheng Xiong (2014) Vps35 haploinsufficiency results in degenerative-like deficit in mouse retinal ganglion neurons and impairment of optic nerve injury-induced gliosis. Molecular Brain, PMID: 24512632.

Hence, the authors of the present manuscript should address this discrepancy and validate that their results unequivocally depend on VPS35 deletion in rod photoreceptors.

Reply: We respectfully argue that the previous x-gal reporter studies ¹ do not undermine the rod's expression of VPS35 and genetic evidence supporting VPS35's role in rods presented in this paper for the following reasons. (i) Our retinal staining convincingly showed VPS35 is expressed in rods and its expression was prominently diminished in conditional knockout (KO) mouse retinas (Fig. 1C, D). This is the gold standard for demonstrating the cell-type specific expression of a given protein, the immune specificity of the antibody used, and the cell-type specific gene deletion. Also, rod's expression of VPS35 was independently demonstrated by the scRNAseq data ^{9, 10}. (ii) The VPS35-floxed mouse line ^{2, 3, 4} and the rod-specific Cre line ^{5, 6, 7, 8} we used to generate the Rod ^{Δ Vps35} mouse line have been widely used. The latter has been used to selectively delete ~40 genes in rods and rod specificity has been tested in each of these papers using a retinal staining method like ours. We have now added the staining of Cre recombinase (revised Fig. S1a) to further show the rod specificity of this mouse Cre line. (iii) Fig. 1c and the scRNAseq data ^{9, 10} showed that retinal ganglion cells expressed very high levels of VPS35. This may partially contribute to the "ganglion-exclusive (or predominant?)" x-gal staining pattern seen in the mouse line that had β -galactosidase knock-in under the Vps35

promoter by Liu et al.,¹. The discrepancy between the scRNA/immunostaining results from the β -gal reporter mouse line is not uncommon in the literature. One explanation is the altered gene regulation caused by the β -gal reporter insertion.

In summary, the following new data/discussion have been added to improve the presentation:

- (i) Photoreceptor nuclear layer-specific expression of Cre-recombinase in KO mice (Fig. S1a).
- (ii) Quantification of retinal staining of VPS35 showed the specific loss of VPS5 in the photoreceptor IS layer (revised Fig. 1d).
- (iii) Persistent rod loss of VPS35 in older KO mice (Fig. 1c, right panel). Since rods contribute to the majority (~97%) of total photoreceptors¹¹, the residual IS signal of VPS35 in the KO was derived from the morphological characteristic cones.
- (iv) Comments about the discrepancy between the VPS35 staining and the x-gal staining in the previous β -gal reporter mouse line.

2) The depletion of rod-specific VPS35 is only shown in 3-week-old mice and only in the inner segment (IS) of the photoreceptors. To help validate their model, the authors should validate the implications of VPS35 depletion at all time points (1, 3, 6 and 9 months).

Reply: Cre-mediated-floxed allele recombination is known to be irreversible. Rod-specific Cre expression has been previously shown to be activated at postnatal 4-7 days⁴⁵ (mentioned in Results). Our original purpose of showing VPS35 staining in 3-week-old mice is to highlight the early Cre activation in the KO. In the revised manuscript, we added the VPS35 staining in 3-month-old KO mice (Fig. 1c, rightmost panel). We did not think the VPS35 staining in 6- and 9-month-old mice is informative because at this time point rods had already shown severe structural disruptions and cell death. The progressively worsening phenotype in the older KO mice, however, independently supports the persistent Vps35 gene deletion.

3) The depletion is only shown in the inner segment of the photoreceptors. However, there is still substantial VPS35 immunoreactivity in the ONL and OPL, likely from the photoreceptors. It would be helpful to quantify VPS35 in these layers to fully evaluate this model.

Reply: As suggested, we have added the quantification of VPS35 signals distributed in different retinal layers (newly added Fig. 1d). We used the disappearance of the inner segment signal of VPS35 to demonstrate the Vps35 gene deletion in rods for the following reasons. VPS35 is undetectable in the OS. Instead, like other early endosomal protein EEA1 (Fig. 3d). VPS35 is primarily localized in the inner segments. The inner segment region is the only retinal layer where no other cell types are intercepted.

It is inappropriate to use the ONL and OPL staining of VPS35 to demonstrate the rod-specific loss because the ONL, while having the photoreceptor nuclear concentrated, also has the radially projected Müller glial processes. VPS35's signal in ONL is largely contributed by its expression in the Müller processes. The OPL is where the photoreceptor terminals, the terminals of several other types of retinal neurons (cones, bipolar cells, and horizontal cells), and the lateral processes of Müller glia are organized. It is impossible to distinguish the contribution of VPS35 from these cells in the OPL.

4) Rod and cone functions were evaluated by scotopic and photopic ERGs, respectively (Figure 2D and E). Surprisingly, photopic ERG response decreased in the VPS35-depleted mice. However, this model should not affect cone photoreceptors. The authors should discuss this result as it could signal some non-specific effects.

Reply: We thank the reviewer for pointing out this seemingly counterintuitive, but unsurprising result. As mentioned above, the cone expression of VPS35 remains in KO (revised Fig. 1c). We do not think the reduced photopic ERG response in our rod-specific VPS35 KO mice is a non-specific event, rather, cone vision defect is secondary because cones depend on rods to live and optimally transduce light (for review²⁹). Rods and cones are electrically and metabolically coupled²⁹. The former are coupled to each other through connexin 36-containing gap junction⁸. These couplings constitute the second scotopic pathway in the retina and are important for twilight vision when both rod and cone visions are involved. Also, rods maintain metabolic fitness by releasing important factors to boost aerobic glycolysis in cones. It is well known that rod-dominant retinitis pigmentosa (RP) patients also lose cone-mediated central vision, and many RP mouse models feature both rod and cone ERG deficits. Similar comments were added in revised DISCUSSION.

5) The microglial activation in response to VPS35 depletion is an interesting finding of the study as it shows the normal microglial response (no depletion of VPS35 in these cells) to lipofuscin or P- α -Syn, and yet, not much was discussed by the authors. Authors ought to refer to retinal microglia activation in the context of AD and retinal degeneration, which has been well characterized. Recommended citations:

- Ni Jin, et al (2021) Front Cell Dev Biol, PMID: 34966736.
- Shi H, et al. (2022) Acta Neuropathol Commun. PMID: 36076283.
- Xu QA, et al (2022). Acta Neuropathol Commun, PMID: 36199154.
- Koronyo Y, et al. (2023) Acta Neuropathol, PMID: 36773106.

Reply: We agreed with this reviewer that the microglial activation in response to the rod-specific loss of VPS35 is rather intriguing. In the revised DISCUSSION we added additional discussion about the possible activation of microglia by the rod-derived wastes (e.g., P- α Syn positive lipofuscins) and the recommended references.

6) The authors should also discuss their findings regarding the location (retinal layer) of P- α Syn in the post-mortem retina of patients with PD and the relevance of their model, characterized by Lewy bodies-like inclusions in and degeneration of photoreceptors.

Reply: To address this important comment, we added the retinal section staining of P- α Syn and its layer-dependent quantification (revised Figs. 5a,b). These studies showed the P- α Syn lesions, originally derived from the VPS35-KO rod terminals, were propagated and widely spread throughout the entire retinal layers including ganglion cells. Previous P- α Syn+ staining of post-mortem human eyes primarily detected P- α Syn-labeled inclusions and neurites in the ganglion cell layer⁴⁰. This report did not, however, rule out the presence of the inclusions in different retinal layers due to technical reasons. i.e., These studies were carried out with retinal whole mounts using the coulometric immunohistochemistry method and observed by light microscope. So the signals in the ganglion cells were more likely to be detected due to the thickness of the tissues⁴⁰. Despite the technical difference, the presence of P- α Syn lesions in the ganglion cells is consistent in these two PD models.

Also, in the revised DISCUSSION, we comment on the relevance of our novel mouse line in modeling α Syn pathology regarding the genesis of Lewy bodies-like inclusion and relevance to various retinal degeneration diseases.

7) The immunoprecipitation method and interaction data are sufficiently described. The authors should detail what kind of interactions are expected. Direct interactions with VPS35? Complexes

containing VPS35? Are interactions detected on the entire VPS35 protein? N-terminal? C-terminal? Some of the interactors do not seem to be specific. For example, numerous histones. The authors should mention these results.

8) Even though the authors focused on HSC70 (HSPA8), it would be interesting to perform enrichment analysis of the interactors detected for VPS35. It could give additional support of the results regarding pathways that involved VPS35.

Reply to (7) and (8): As suggested, we validated the physical interaction between VPS35 and HSC70 using the pull-down assays of recombinant proteins. These experiments showed that HSC70 (his-tagged) bound to the VPS35 (GST-tagged) primarily through its C-terminal half (revised Fig. 6d).

We also added the enrichment pathway analyses of VPS35-interactomes using DAVID, Fisher's and GSEA methods (Fig. 6a-c, Supplementary table 2-4). These studies unbiasedly identified several pathways that have previously linked to VPS35 (e.g., PD, AD, endocytosis, synaptic recycling), and hence, validated the assays. These findings also identified several pathways that have not yet been previously recognized for VPS35 (proteasome, protein (re)folding, protein processing at ER, supramolecular polymer) but have been linked to protein quality control.

In general, one cannot tell whether every individual molecule isolated from the immunoprecipitation experiments is through specific interactions. Many histone proteins identified among the VPS35 interacting protein isolates are known as HSC70 binding proteins (e.g., Histone H3.1^{41, 42, 43}, histone H2A⁴⁴) and are found in several enriched pathways (e.g., RNA binding, protein containing complex binding, cadherin binding, cell adhesion molecular binding). Therefore, we decided not to call out any particular isolates as non-specific hits without future experimentally validation.

Minor points

- 1) The legend of Figure 2D and E is difficult to understand. The authors should refer to the bar graphs or the traces and better explain a- and b-waves.
- 2) In the legend of Figure 7 (page 34, line 1084), macrophagic should be autophagic.
- 3) In the legend of Figure S2 (page 37, line 1101), (C, D) should be (B, C).
- 4) The supplementary table 1 is incomplete (page 40). Please add the remaining interactor proteins after HSPA5.
- 5) On page 40, line 1123 and on page 41, line 1124, please correct as Supplementary Table 1 and Supplementary Table 2, respectively.

Reply: We have made editorial changes of the above notes accordingly.

Reviewer #2 (Remarks to the Author):

Reply: Thank you for your service.

Reviewer #3 (Remarks to the Author):

The results are interesting and relevant not only to Parkinson's disease. The relevance of the discovered VPS35-HSC70-pasyn interaction are very interesting; its impact could be further explored by testing it in different pathological contexts, such as alpha-synuclein overexpressing models.

The main problems are related to methodological issues, which make it difficult to evaluate the robustness of the presented data and the lack of in vivo evidence of visual acuity impairment in these animals. In particular:

Q1)- Single-point histograms should be reported.

- Figure 1E and 1F: Is there any evidence in the literature, with mice of the same genetic background as this study, that the thickness of the ONL and OPL does not change in the 6- and 9-month-old control animals? Note that the scotopic ERG, in control animals, decreases already at 6 months compared with younger animals (Fig. 2D). An age-matched control should be used in all experiments.

Reply: As suggested, the ONL thickness of aged-matched (and genetic background-matched) controls and KO were provided in revised Fig. 1e, f, S1b, and S1c. The ERG data of aged-matched controls and KO was shown in the original manuscript (now revised Fig. 2a,b). The age-dependent ERG signal decline has been reported in wild type mice^{46, 47, 48}.

Q2)- Immunofluorescence analyses are often not quantified by standard stereological methods; it is necessary to have them.

- Co-localization experiments (Fig. 3E) must be quantified, using unbiased counting methods (stereology).

Reply: As suggested, we added unbiased quantification (see Methods) for all immunostaining shown (revised Figs. 1d, 2c, 3e, 3f, 5b, 5f, S6a, and S6a). We also quantified the colocalization of LC3 and Lamp1 (Fig. 3H), as well as the colocalization of P- α Syn and autofluorescence (Fig. 7g) by presenting the Pearson coefficient (described in revised Results).

Q3)- Figure 3D is missing images of the immunoblots. All original blots should be provided; I could not find them in the source data.

Reply: We have added the representative immunoblots of the original Fig. 3d (revised Fig. 3g). All original blots were presented in the revised Fig. S9 and Fig. S10.

Q4)- Analysis of p62 or other substrates of autophagy should be provided to verify impairment of autophagy. In general autophagosome/lysosome fusion needs to be evaluated using appropriate assays (PMID: 33634751).

Reply: A plethora of literature has comprehensively demonstrated that VPS35 deficiency and mutation leads to impaired autophagy flux using the suggested assays^{14, 15, 16, 17, 18, 19, 20}. Unfortunately, not all the gold standard assays for measuring autophagosome/lysosome fusion are feasible in vivo, particularly when a single cell has been genetically manipulated in the multicellular tissue. In our case, this pertains to the rod-specific deletion in the retina. As a result, we did not detect significant changes of P62 expression in the KO retinas on immunoblots.

Q5) - The data on oligomeric alpha-syn are not convincing and are not quantified. Similarly, the

IF for p-asyn has not been quantified. To confirm the presence of aggregates, proteinase-K resistance should be performed and evaluated.

Reply: We have replaced the original immunoblots with better represented ones with their quantification to demonstrate the pattern in the increase of the oligomerization of P- α Syn in KO mouse retinal lysate (in Fig. 5e,f). We add the quantification of the retina-stained P- α Syn (revised Fig. 5a,b). We add the data to show that KO retina expressed P- α Syn is resistant to proteinase-K treatment (revised Fig 7h).

Q6) - Control images for 5E and 5H are not reported.

Reply: The original Fig. 5e show the colocalization of endogenous HSC70 and EEA1 expressed in 661W cells. For the control of Fig. 5e, since EEA1 antibody has been extensively validated, we tested the immunospecificity of HSC70 antibody using two methods. First, we showed little or no staining signal when HSC70 antibody was avoided (not shown). Second, we showed that the HSC70 staining was evidently reduced in 661W cells had HSC70 suppressed (via transfection of HSC70-shRNA-IRES-GFP (revised Fig. 5b).

We added the control for the original Fig. 5h (top panel of revised Fig. 6i). Specially, we showed, unlike the VPS35-KD, the control cells did not exhibit any prominent α Syn and/or P- α Syn aggregates in Myc- α Syn transfected cells.

Q7) - Although the ERG provides an indication that KO mice might have a visual acuity defect, specific behavioral tasks (using an appropriate number of animals per group based on power analysis) are needed at 1 and 3 months to assess functional changes in the observed defects.

Reply: As suggested, we added the scotopic visual acuity test results of 2-3-month-old mice in revised Fig. S4b. The KO (vs. Ctrl) showed lower visual acuity (cycles/degree; N=13 (Ctrl) N=6 (KO)), though the difference is not statistically significant. These results are expected because, unlike ERG which measures the electrical response of the retina to light, the plasticity of brain may adapt the retinal-mediated light transduction defect; this reflects on the visual acuity behavioral task. Mice of young age do not perform consistently in this behavior test so we did not perform the visual acuity test in 1-month-old mice.

Reviewer #4 (Remarks to the Author):

The manuscript entitled " Mutant mice with rod-specific VPS35 deletion exhibit visual deficit, synaptopathy, and retinal synucleinopathy" written by Fu et al investigated the role of rod-specific VPS35 in retinal synaptopathy, synucleinopathy and visual function. They found that rod-specific VPS35 deletion causes dysregulated OS genesis, endolysosomal homeostasis, protein degradation and autophagic flux in rods as well as early rod terminal loss before rod cell death using both light microscopic and electron microscopic analyses. Moreover, proteomic analysis demonstrated a crosstalk between VPS35 and chaperon HSC70. The findings are interesting. However, the major issue in the manuscript is that some of conclusions made by the authors cannot be backed up by the current data but by speculations.

Q1. 661W cell line is cone photoreceptor cells. However, the authors used the cell line for rod cells. Even though VPS35 signals appeared a fraction of 661W cells (Fig. 5E), the authors still used the cell line to study VPS35-HSC70 interactions and claimed to uncover a novel crosstalk between VPS35 and HSC70.

Reply: 661W was initially derived from retinal tumors expressing SV40 T-antigen in photoreceptors of mice. This cell line expresses both cone-specific³⁰ and rod-precursor^{31, 32} proteins and has widely been used as a photoreceptor-like line to study the rod-dominant retinal degeneration disease RP^{33, 34, 35, 36, 37, 38, 39}. There is no other characterized photoreceptor cell line, we argue our usage of 661W cell lines is reasonable.

Q2. For proteomics, the authors didn't even perform KEGG pathway analysis and protein-protein interaction analysis and claimed to identify HSC70 as a VPS35-interacting protein.

Reply: As suggested, we used the recombinant protein pull-down assays to validate the physical interaction between VPS35 and HSC70 (revised Fig. 6d; also see Reviewer 1, Q7/8).

We also presented the enrichment pathway analyses (revised Fig. 6a-c, Supplementary Tables 2-4) to show that the VPS35-interacting proteins are overrepresented in the pathways of synucleinopathy (e.g., PD). The pathways are consistent with the observations described in this paper (e.g., synaptic vesicle recycling), and the pathways previously linked to α Syn lesions (e.g., Prion disease, proteasome, protein refolding, protein processing in ER, supermolecule polymers). This finding unbiasedly connects VPS35 to several pathways previously linked to α Syn pathology and HSC70's chaperon function.

Q3. The authors generated a mutant mouse line with rod-specific VPS35 deletion and studied the role of rod-specific VPS35 in retinal synaptopathy, synucleinopathy and visual function. Yet, the authors claimed that fundus AF dots in the mutant mouse can be used as a potential early PD biomarker.

Reply: Our results (Figs 7 and 8) showed P- α Syn is closely associated with the lipofuscins that had autofluorescent (AF) property. Concentrated P- α Syn-lipofuscin lesions expressed in the retina can be detected by fluorescent funduscopy in real-time. Therefore, we propose the unusual fundus AF foci should be explored as a non-invasive biomarker in PD as it is a good proxy for the P- α Syn lesions. We did not directly claim that fundus AF dots in the mutant mouse can be used as a potential early PD biomarker.

4. The authors stated that VPS35 is a hub that functionally couples the three major pathways endolysosomal trafficking, protein quality control, and selective microautophagy. The three major pathways were identified solely by imaging analyses. There is no any functional study to confirm the relationship between VPS35 and the three major pathways.

Reply: A plethora of literature has extensively demonstrated VPS35's functional role in endolysosomal trafficking and selective autophagic pathways using cell culture systems^{12,13 14, 15, 16, 17, 18, 19, 20}. Here, we used several independent approaches – biochemical (Fig 3C 3G), immunohistocal (Fig. 2c, d, 3d-f, 3h), morphologically (Fig. 3a, 4a-g), and electrophysical (Fig. 2a, b) - to comprehensively show VPS35's role in endocytic trafficking (hence, synaptic recycling, endomembrane homeostasis) and autophagic pathways in rods in vivo.

HSC70 has well-recognized roles in protein quality control and selective autophagy^{21, 22, 23, 24, 25, 26, 27, 28}. The novel VPS35-HSC70 interaction was first implicated by proteomics studies. We validated this by using pull-down assays with recombinant proteins (revised Fig. 6d), colocalization studies (Fig. 6e), and functional interaction in α Syn aggregation in cell cultures (Fig. 6g-i). Furthermore, the bioinformatics pathway analyses identified "protein (re) folding" as

an enriched pathway overrepresented in the VPS35-interacting proteins (Fig. 6a, c). Also, the newly added pathway analyses showed “Protein (re) folding” and “protein processing in ER pathways” were overrepresented in the VPS35-immunoprecipitated proteins (revised Fig. 6a-c). These datasets, taken together, are consistent with the notion that VPS35 is the node of the three major pathways (endolysosomal trafficking, protein quality control, and selective microautophagy) involved in α Syn lesions. With that being said, we did tone down the claim by removing this exact sentence from the last part of INTRODUCTION. We included this comment in the revised DISCUSSION with modified wording.

Reviewer #5 (Remarks to the Author):

In the current study, Fu et al. generated a rod-specific VPS35 knockout mouse model and addressed synucleinopathy in the eye of the animal, exhibited by microglia activation with p-a-syn and autofluorescence accumulation, which the authors suggested as a potential biomarker for PD. In addition, the authors suggested presence of VPS35-HSC70 cross talk via late endosome sequestering and a-syn aggregation. This study takes a novel approach and is very interesting in both study design and considering eye degeneration has not been studied much in the field of synucleinopathies. However, there are some important concerns that currently diminish the study.

Q1) Due to the fact that VPS35 is strongly associated with autophagy, the dysregulation of autophagy, cellular toxicity, and accumulation of a-syn aggregates are expected. The manuscript appears to this reviewer to be the sole discoverer of the phenomenon. Since the authors claimed this model may apply to synucleinopathies, the authors should include data which can show the relation between this model and synucleinopathies. For example, can the authors detect a-synuclein neuropathology in the brains of aged KO mice (at least in the visual cortex)?

Reply: With due respect, investigating the P- α Syn pathology in the brain, which requests careful data curation, requires more time and is beyond the current scope. We added several pieces of new data to further support that this new model features a spectrum of cellular and pathological phenotypes previously linked with synucleinopathies. These mouse retinas exhibit early synapse loss (Fig. 2), neuronal cell death (revised Figs. 1e,f and S1b,c), and microglia activation (revised Figs. 7a-d and Figs S6a-c). They also feature the pathological hallmarks: LB-like inclusions (revised Fig. 5c, 5d, 7i) and P- α Syn aggregation (revised Figs. 5e,f, 7g). Moreover, several lines of evidence strongly support the propagation of P- α Syn pathology and its spreading from the origin of the lesion (VPS35-KO rods). (i) P- α Syn staining was broadly increased throughout the entire retina (Fig. 5a, b) including the ganglion cells, which extend axons (optical nerves) reaching into the brain. (ii) The spreading of P- α Syn-labeled LB-like inclusions to the microglia (Fig. 7g-i). (iii) The spreading of the α Syn lesions has been often analogous to that of the prion. Prion disease is one of the pathways enriched by the dataset of VPS35-interacting proteins (Fig. 6a). These data, taken together, consistently indicate the potential of this new mouse line for modeling synucleinopathies. We believe this “retina-focus” report is a complete story within its own right.

Q2) Does the synucleinopathy animal model also show retinal degeneration? Or autofluorescence in the microglia? Is the degeneration inhibited by an a-syn knockout?

Reply: Mammadova et al.,⁴⁹ studied the postmortem retinal sections of transgenic mice overexpressing α Syn A53T mutant (TgM83) and injected with brain homogenates of clinically ill

mice. These mice developed retinal phenotypes several months before clinical illness (12-16-months-old); they had retinal thinning, increased retinal expression of P- α Syn and increase in subretinal microglia. They did not, however, investigate whether these microglia express autofluorescence, nor whether the deficiency of α Syn may rescue the degeneration phenotype. Investigating whether depleting α Syn might reduce the retinal degeneration in the current mouse model is a question of interest, but outside the current scope.

Q3) The authors claimed that there is microglia activation through infiltration into the retinae. However, microglial activation should be determined in multiple ways, such as with inflammatory cytokine gene expressions and morphological differences.

Reply: This is an excellent suggestion. In addition to the original demonstration of the infiltrated microglia expressed disease-associated microglia (DAM) markers (revised Fig. 7c), we added several pieces of additional data to support the microglial activation. (i) Morphological and morphometric analyses to determine microglial shape/ramification (newly added Fig. 7b, S5a, b) and (ii) Quantitative immunostaining of inflammation markers IL-1 β and CD68 (newly added Figs. S5a, c, d). These data show that infiltrated subretinal KO microglia are highly activated and more activated compared to the microglia distributed in the OPL and IPL of either KO or Ctrl.

Q3) Where was a-syn was expressed in the retina? Microglia don't express a-synuclein, so internalized and accumulated p-a-syn should be exogenously internalized into the microglia. However, significant a-syn pathology was not observed in the retina. As such, the source of microglial p-a-syn should be addressed. VPS35 is also associated with a-syn degradation. However, if the rods express a-syn, why did the VPS35 knockout not induce a-syn accumulation in the rod?

Reply: "Where was a-syn was expressed in the retina?...if the rods express a-syn, why did the VPS35 knockout not induce a-syn accumulation in the rod?" In the revised Fig. S4a, we showed the retinal staining of α Syn; brightly stained α Syn-labeled inclusions were predominantly appeared in the rod photoreceptor layer of KO (arrows, Fig. S4a). Similar P- α Syn-labeled inclusions were also primarily expressed in the photoreceptor layer of KO (revised Fig. 5c). In the revised DISCUSSION, we added comments explaining why the LB-like inclusions are mainly restricted in the photoreceptor layer despite the broad retinal expression of α Syn (Fig. S4a) and P- α Syn (Figs. 5a, b).

"Significant a-syn pathology was not observed in the retina?" In the original manuscript, we demonstrated α Syn pathology by showing the emergence of prominent P- α Syn inclusions in KO at both light microscopic (Fig. 5c) and EM (Fig. 5d) levels. We have now provided new data to show the increase of TX-100 insoluble (revised Figs. 5e,f) and proteinase K-resistant (Fig. 7h) P- α Syn aggregates in KO. Furthermore, the newly added Figs. 5AB show a broad increase of P- α Syn throughout the KO retina, indicating the propagation and spread of α Syn lesion in these retinas.

"The source of microglial p-a-syn should be addressed" Our data showed (i) P- α Syn aggregates were accumulated in late endosomes (LEs) in VPS35-KD cells (Fig. 6g-i), (ii) lipofuscins are the lipid aggregates deposited from LEs (Fig. 4d,e), (iii) Lipofuscins escaped from VPS35-KO rod terminals were engulfed by neighboring microglia (Fig. 4f, g, 7e, S3b, c) which subsequently migrated to the subretinal region (Fig. 7e). (iv) P- α Syn was frequently associated with lipofuscins (Fig. 5d, 7g-i). These results collectively support a model (depicted

in Fig. 8h) that P- α Syn-lipofuscin aggregates, derived from the LEs of VPS35-KO rod terminals, are the primary source of microglia-expressed P- α Syn.

Q4) Other critiques:

Introduction

Line 52 – The optical symptoms of PD is not a very well-known topic in the field. Therefore, more detailed information to introduce the topic should be included, such as the % of occurrence and visual disturbance symptoms (color, contrast defect, etc.).

Reply: We added the requested information in both revised INTRODUCTION and DISCUSSION. Briefly, around 80% of PD patients exhibit at least one of several possible visual symptoms including reduced visual acuity, spatial contrast sensitivity, depth perception, visual hallucination, and color vision^{50, 51, 52}.

Line 56 – P129, not P219 and please use same abbreviation for a-syn (Ex, a-Syn or P-aSyn).

Reply: We have corrected these typos.

Line 62 – Pathogenic interaction between HSC70 and a-syn has been reported, which should be mentioned.

Reply: As suggested, we included references describing the pathogenic interaction between HSC70 and α Syn in revised Introduction and Results.

Results

Fig 1a,b – Provide magnification and size of picture. Provide rationale for why Ribeye, snap25, and GluR6 were analyzed in the mice.

Reply: We added a scale bar in Fig. 1a. Fig. 1b is a schematic diagram, not true to scale. We added the rationale for analyzing Ribeye SNAP25, and GluR6 in revised Results.

Fig 1c – Although immunostaining provided visual quantification data, a more solid and conservative analysis such as an immunoblot is required to validate the knockout of VPS35.

Reply: With due respect, immunoblotting of total retinal lysates won't accurately reflect rod-specific loss of VPS35 because VPS35 is widely expressed in many retinal cell types. In fact, both our immunostaining results (revised Fig. 1c, 1d) and public-available scRNA datasets^{9, 10} showed that several retinal cell types express more abundant VPS35 than rods. Immunostaining assay, as we showed in Fig. 1d and revised Fig. 1d, has been considered a gold standard for demonstrating rod-specific gene depletion^{5, 6, 7, 8}.

Fig 1d – Provide age-matched control HE staining in parallel with KO mice, which will address whether degeneration is age-related and/or progressive.

Reply: As suggested, we showed age-matched, HE-stained retinal sections of both control and KO and the ONL thinning quantification to demonstrate the progressive retinal degeneration (revised Fig. 1e and 1f).

Fig 2c – According to Fig 1, control and tg did not show much difference in at one month. To this reviewer, data analysis for mice that are 3 months old or older would more appropriate. In

addition, this reviewer also could indicate short and dark tube-like shape in the WT, which needs to be addressed.

Reply: The reviewer asked why we showed the synapses of 1 month-old KO (original Fig. 2c, revised Fig. 2e) rather than 3-mon-old KO. In fact, we have already provided the synaptic morphology of both ages (1-mon-olds in Fig. 2e; 3-mon-olds in Fig. 4). The purpose of showing young mice is to depict the early ultrastructural changes that might be unobservable by the immune-histological examinations.

Also, we enlarged the electron micrographs (original Fig. 2c) and clearly marked the short dark tubules (now revised Fig. 2e).

Line 131 – Again, many scientists in the field are not familiar with the electroretinogram, so please provide a brief explanation of the method.

Reply: We added the description for electroretinogram (ERG) in revised Methods and Results.

Fig S1a – Provide age-matched analysis data

Reply: We provide the age-matched (i.e., OS length) dataset of the original Fig S1a in revised Fig. S2a.

Fig 3a,b – Why was OS analyzed in 1 month old mice and retinal staining done for 3 month old mice?

Reply: In fact, the EM images of both 1-mon-old and 3-mon-old OS were shown in the original manuscript in Fig. 3a, b and 4a, respectively. The former highlighted the early ultrastructural changes. The immunostaining performed in 1- and 3-mon-olds obtained similar results; we showed the representative images.

Fig 3d – Provide original image of western blot

Reply: The original images of all immunoblots are now included in revised Figs. S9 and S10.

Fig 3e – Data looks clear, but again control data should be provided.

Reply: We added the control images in the original Fig. 3e (revised Fig. 3h).

Fig 3g – Western blot should be analyzed in the same blot to avoid unnecessary misunderstanding.

Reply: We agreed. The revised Fig. 3c (replaced the original Fig. 3g) showed the blots with all three controls and KO samples run on the same gels alongside the internal housekeeping gene control tubulin.

Fig 3h – Quantification of western blot is difficult to understand. This should be transformed as it is in Fig 3d and include individual and control data.

Reply: We changed the format of the original Fig. 3h (now revised Fig. 3c) as suggested format.

Lines 164-5 - Are these proteins targeted by the proteasome or autophagy? It would be

beneficial to the paper if this can be validated in an in vitro model system.

Reply: We respectfully defer the proposed experiment because no cell line endogenously expresses any of our tested photoreceptor-specific proteins (e.g., rhodopsin, peripherin2/rds, IRBP). Testing overexpressed transfected proteins in cell cultures is not physiologically relevant. Furthermore, the degradation of photoreceptor protein is known to involve multiple cell types surrounding them (e.g., RPE, microglia and Müller glia). These cells have robust phagocytic activity and engulf the materials released from photoreceptors during both physiological repair and pathological conditions. No in vitro model system can recapitulate this complex intercellular interaction.

Fig. 4 – Again, the FIB-SIM data of the age-matched control should be included. Appropriate quantification should also be added; for example, the numbers of SV or MLB.

Reply: As suggested, we added a FIB-SEM image of an age-matched (3-month-old) control rod terminal (revised Fig. S3a). The three different types of wastes - MLB (multilaminar body), lipofuscin, amorphous aggregate - observed in VPS35-KO rod terminals were not detectable in the control rod terminals. We added the quantification and respective distribution of these wastes in the KO rod terminals, Müller glia and microglia cells (Fig. S3b, c). These data provide visual proof of cell-to-cell transfer of rod-derived pathology. We did not count the number of synaptic vesicles (SVs) due to technical issues. We believe that the quantitative immunostaining of the synaptic ribbon (i.e., active zone) marker Ribeye (Figs. 2c,d) is a proximity of abnormal SV docking.

Fig. 5 – Why were 3-month-old animals used for immunofluorescence staining and 6-month-old animals for immunoblot analysis? B and C seems to be KO, but were not labeled and need controls. Where does p-syn accumulate? In which cells? And how about a-syn expression?

Reply: In the revised manuscript, we showed the immunostaining (Fig. 5a-d, S4a) and immunoblots (Fig. 5e, f) of α Syn (Fig. S4a) and P- α Syn (Fig. 5a-d) of 3-month-olds. We removed the data previously obtained from 6-month-olds to be consistent.

Regarding “Where does p-syn accumulate? In which cells? And how about a-syn expression?”

Reply: Please also see reply of Q3 above. Briefly, we added the retinal staining of P- α Syn (revised Fig. 5a, b) and α Syn (revised Fig. S4a). Accumulated α Syn and P- α Syn in LB-like inclusions were prominently detected in the inner segment-to-outer segment region of photoreceptor layers. The P- α Syn inclusions were also abundantly detected in subretinal microglia (Fig. 7g, h).

Fig. 6 – Immunolabeling data seems reliable, but flat mount does not. To this reviewer, all signals might originate from AF structure, so would suggest double labeling Iba1 with p-syn instead of the flat mount.

Reply: With due respect, immunoblotting of total retinal lysates is unable to accurately reflect the rod-specific loss of VPS35 because VPS35 is also expressed in other retinal cell types. This is especially problematic because of our immunostaining results (Fig. 1c) and scRNA data^{9, 10} that showed several other retinal cell types express more abundant VPS35 than rods. We respectfully disagree with the comment that the immunostained P- α Syn signals originate from AF in flat mount data because (i) the fluorescence of AF granules (peak excitation/493nm/emission 517 nm) did not overlap with the Alexa 647 dyes (653 nm

excitation/680 nm emission) which was used to label P- α Syn (see revised RESULTS), (ii) the shape of P- α Syn aggregates and AF granules are not completely identical, and (iii) our colocalization studies show the Pearson's correlation coefficient of these two structures is ~0.5-0.6, i.e., about 50-60% signals are superimposable (see RESULTS). Note that if the P- α Syn was derived from the AF bleed through, the Pearson's correlation coefficient would be close to 100% instead.

The main purpose of this flat mount image is to have the spatial resolution to highlight the strong overlap between P- α Syn and AF, so we can utilize the fundus AF dots to detect P- α Syn lesions in live mice.

Fig 7b – Please include age-matched control data.

Reply: As suggested, we have added the age-matched control mouse fundus images (original Fig. 7b, now revised Fig. 8a).

Discussion

What are the benefits and limitations of current model compared to well-known synucleinopathy model? The relation and application of the authors findings with synucleinopathy should be addressed.

Reply: We have addressed these important suggestions vigorously in the revised DISCUSSION.

REVIEWER COMMENTS

Reviewer #1 (Remarks to the Author):

The authors responded to most of the referees' comments and improved the clarity and validity of their findings in this manuscript. However, there remain several concerns that need to be addressed:

1. Co-immunostaining of rod-specific photoreceptor marker (i.e. Rhodopsin) with VPS35 to demonstrate the specific VPS35 expression within IS rod photoreceptors is required, alongside demonstrating its depletion in the rods of KO mice as compared to control mice. This is important considering previous reports showing VPS35 expression only in the RGCs.
2. The authors provide two references of scRNAseq data 9, 10 (references 74, 75 in the manuscript) to provide evidence of VPS35 expression in photoreceptors. However, these two studies were done in the human retina and these results should be validated to mice. Also according to the scRNAseq data 75 at Cell-type gene expression browser - adult human retina (shinyapps.io), VPS35 RNA was detected in less than 10% of rod photoreceptors and at very low levels in Müller glia. Authors should mention these limitations in the Discussion.
3. A quantification of co-immunostaining of VPS35 signal with Ribeye (specific to synaptic terminals of rod photoreceptors) to demonstrate potential changes in VPS35 expression in rod synaptic terminals in KO vs controls is highly recommended.
4. Regarding the potential of no change in VPS35 expression in the ONL or OPL, authors are encouraged to co-stain for Müller glia marker(s) and quantify the rod specific VPS35 in these layers, to further validate their findings in the targeted conditional KO model.
5. All bar graphs should include the individual data points to allow the readers a careful assessment of sample size and data distribution for each group.
6. In page 3, line 120-123: "...its rod-specificity has been extensively characterized 5, 6, 7, 8." and "As rods contribute to the majority of total photoreceptors 11..." - The reference numbers are incorrect. Authors should add the references to the list and provide the correct numbers.
7. Fig. 1c: the representative microscopic image for the KO retina (left bottom) is missing the GC/NF layers. Please include.
8. Authors should specify the age of the mice in all figures or figure legends.
9. In page 8, line 378-380: "The decline of both rod and cone ERG is commonly observed in RP patients and many RP mouse models even though rod is a primary lesion site." Please provide the appropriate references; citation #77 is not relevant to the above statement.
10. What is the earliest age that retinal P- α Syn can be detected and is elevated in the VPS35-KO mice? Is the ONL neurodegeneration a direct consequence of rod VPS35 depletion or a consequence of P- α Syn accumulation?
11. In page 3, line 120-121: "However, the retinal thinning and the detection of P- α Syn inclusions/neurites in postmortem PD patient eyes 12 ", please add "...inclusions/neurites of RGCs in postmortem..." to match the specific findings of P- α Syn in RGC neurites in ref #12.

Reviewer #2 (Remarks to the Author):

Reviewer #3 (Remarks to the Author):

General comment:

Single dots graphs should be used in all figures

Point by point to address:

R2 to Q2: As suggested, we added unbiased quantification (see Methods) for all immunostaining shown (revised Figs. 1d, 2c, 3e, 3f, 5b, 5f, S6a, and S6a). We also quantified the colocalization of LC3 and Lamp1 (Fig. 3H), as well as the colocalization of P- α Syn and autofluorescence (Fig. 7g) by presenting the Pearson coefficient (described in revised Results).

To revise: 3H reports photographs of the colocalization, but not quantifications. Please, add the graph

R5 to Q5) - We have replaced the original immunoblots with better represented ones with their quantification to demonstrate the pattern in the increase of the oligomerization of P- α Syn in KO mouse retinal lysate (in Fig. 5e,f). We add the quantification of the retina-stained P- α Syn (revised Fig. 5a,b). We add the data to show that KO retina expressed P- α Syn is resistant to proteinase-K treatment (revised Fig 7h).

To revise: Revised 7h shows one cell, which is not sufficient to prove the resistance to pk. Again, quantification is necessary and larger panels for h including more cells should be provided together with the zoom.

Quantification of the oligomer in figure e is really challenging. There is no evidence of the presence of the oligomeric form.

Q7) -R7: As suggested, we added the scotopic visual acuity test results of 2-3-month-old mice in revised Fig. S4b. The KO (vs. Ctrl) showed lower visual acuity (cycles/degree; N=13 (Ctrl) N=6 (KO)), though the difference is not statistically significant.

These results are expected because, unlike ERG which measures the electrical response of the retina to light, the plasticity of brain may adapt the retinal-mediated light transduction defect; this reflects on the visual acuity behavioral task. Mice of young age do not perform consistently in this behavior test so we did not perform the visual acuity test in 1-month-old mice.

To revise: I agree with the authors regarding the interpretation of the results. However, they should be discussed in relation to previous evidence in the literature demonstrating visual acuity deficits in animal models of wild-type alpha-synuclein overexpression in the retina of control mice (Marrocco et al., Scientific Reports, 2020). Additionally, the absence of a visual acuity defect in the model represents a limitation of the study that should be clearly emphasized in the manuscript.

Reviewer #4 (Remarks to the Author):

The authors didn't fully address my early concerns and didn't provide new evidence but new arguments. For instance, the authors reasoned that 661W cell line expresses rod-precursor, there is no other characterized photoreceptor cell line, they thus believe that 661W cell is the only choice available. However, there are three strong arguments against them. One is that what they found on the cell line is irrelevant to the function of VPS35-deficient rods in the mice. Unless, the authors can demonstrate that VPS35-deficient-661W cell and VPS35-deficient rods are the same cells. Secondly, primary rods from VPS35-deficient mice are obviously the best choice for in vitro experiments in this study. However, the authors did not do it. Thirdly, the authors found that rod-specific VPS35 deletion causes dysregulated OS genesis, endolysosomal homeostasis, protein degradation and autophagic flux in rods. As one Reviewer mentioned (and I totally agree) that previous studies have demonstrated that VPS35 is strongly associated with autophagy, the dysregulation of autophagy, cellular toxicity, and accumulation of a-syn aggregates are expected but not unexpected. Therefore, we wonder what are new significant findings in this study.

Reviewer #5 (Remarks to the Author):

The authors addressed this reviewer's requests fairly well, except for the first critique. I still believe that neuropathology analysis of the current model is required to claim this model as a

synucleinopathy model. A model for synucleinopathy has to mimic the neuropathology and symptoms of the disease. Although many PD patients have visual symptoms, it is not yet fully understood whether these originated from retinal dysfunction or brain degeneration. Therefore, the authors should provide suitable neuropathology analysis to use the terms specific to a synucleinopathy model. If the authors can't analyze or determine them, the authors should use a different set of terminology to describe this model.

Abbreviation keys: PD: Parkinson's; P- α Syn; phosphorylated α -Synuclein; OS: outer segment; IS: Inner segment; KO: knockout; ONL: outer nuclear layer; OPL: outer plexiform layer; Ctrl: control. cSLO: confocal scanning laser ophthalmoscope

REVIEWER COMMENTS

Reviewer #1 (Remarks to the Author):

The authors responded to most of the referees' comments and improved the clarity and validity of their findings in this manuscript. However, there remain several concerns that need to be addressed:

1. Co-immunostaining of rod-specific photoreceptor marker (i.e. Rhodopsin) with VPS35 to demonstrate the specific VPS35 expression within IS rod photoreceptors is required, alongside demonstrating its depletion in the rods of KO mice as compared to control mice. This is important considering previous reports showing VPS35 expression only in the RGCs.

Reply: We performed the requested experiments with a minor modification. Because rhodopsin is predominantly expressed in the outer segments, we instead performed co-immunostaining of VPS35 with ATP1A which marks the plasma membrane of the photoreceptor IS. These results revealed the IS expression of VPS35 in ctrl mice and the loss of the majority of IS signal of VPS35 in KO (revised Fig. S1b). We also added the co-staining of VPS35 and cone arrestin, a cone marker (revised Fig. S1c). This shows the remaining VPS35 signal in the IS was from the cones.

2. The authors provide two references of scRNAseq data 9, 10 (references 74, 75 in the manuscript) to provide evidence of VPS35 expression in photoreceptors. However, these two studies were done in the human retina and these results should be validated to mice. Also according to the scRNAseq data 75 at Cell-type gene expression browser - adult human retina (shinyapps.io), VPS35 RNA was detected in less than 10% of rod photoreceptors and at very low levels in Müller glia. Authors should mention these limitations in the Discussion.

Reply: The scRNAseq of mouse retinas showed that, like in humans, VPS35 is expressed in ~10% of rods (vs. ~40% of retinal ganglion cells) (Hoang, Wang et al. 2020). A low-level expression of VPS35 in rods does not rule out its biological importance as we demonstrated using genetic approach. We added the mouse scRNAseq reference and the related comments in the revised Discussion as requested.

3. A quantification of co-immunostaining of VPS35 signal with Ribeye (specific to synaptic terminals of rod photoreceptors) to demonstrate potential changes in VPS35 expression in rod synaptic terminals in KO vs controls is highly recommended.

Reply: As requested, we added the representative images (revised Fig. S1e) and quantification (revised Fig. S1f) of the overlap between VPS35 and Ribeye in control and KO rod synapses. This data showed the colocalization of VPS35 with Ribeye, and such overlap is significantly reduced in KO.

4. Regarding the potential of no change in VPS35 expression in the ONL or OPL, authors are

encouraged to co-stain for Müller glia marker(s) and quantify the rod specific VPS35 in these layers, to further validate their findings in the targeted conditional KO model.

Reply: As requested, we performed the co-staining of VPS35 and glutamine synthase that marks the Müller glia processes spanned across outer nuclear layer (ONL). We added the representative images (Fig. S1c) and quantification (Fig. S1d) demonstrating Müller glia-associated VPS35 signal in ONL region was comparable between Ctrl and KO (revised Fig. S1c, d).

5. All bar graphs should include the individual data points to allow the readers a careful assessment of sample size and data distribution for each group.

Reply: We revised all bar graphs that applicable by including individual data points.

6. In page 3, line 120-123: "...its rod-specificity has been extensively characterized 5, 6, 7, 8." and "As rods contribute to the majority of total photoreceptors 11..." - The reference numbers are incorrect. Authors should add the references to the list and provide the correct numbers.

Reply: Our apologies, we editorially corrected the citation errors.

7. Fig. 1c: the representative microscopic image for the KO retina (left bottom) is missing the GC/NF layers. Please include.

Reply: We replaced a new image set to include the ganglion cell (GC)/ neurofilament (NF) layers of Fig. 1c.

8. Authors should specify the age of the mice in all figures or figure legends.

Reply: We confirmed the ages of the mice used in all figures are specified in the legends.

9. In page 8, line 378-380: "The decline of both rod and cone ERG is commonly observed in RP patients and many RP mouse models even though rod is a primary lesion site." Please provide the appropriate references; citation #77 is not relevant to the above statement.

Reply: We apologize for the confusion. The original citation #77 (revised citation #85) was referred to the sentence "This is because rods and cones are electrically and metabolically coupled". In the revised manuscript, we added references citing the sentence "The decline of both rod and cone ERG is commonly observed in RP patients and many RP mouse models even though rod is a primary lesion site."

10. What is the earliest age that retinal P- α Syn can be detected and is elevated in the VPS35-KO mice? Is the ONL neurodegeneration a direct consequence of rod VPS35 depletion or a consequence of P- α Syn accumulation?

Reply: Fig. 5a-b showed the significant increase of P-aSyn staining of 3-month-olds, a relatively young age. Fig. 1c showed VPS35's depletion in rods was apparent in 3-week-olds. The causes of photoreceptor degeneration in KO mice, while initiated by the rod depletion of Vps35, is likely to be multifactorial. Presenting the earliest time point of P-aSyn increase in KO (must between 3-weeks and 3-months) might not be adequate to claim P-aSyn accumulation is critical for rod cell death. With due respect, we deferred this interesting question to future study.

11. In page 3, line 120-121:” However, the retinal thinning and the detection of P-aSyn inclusions/neurites in postmortem PD patient eyes 12 ”, please add “...inclusions/neurites of RGCs in postmortem...” to match the specific findings of P-aSyn in RGC neurites in ref #12.

Reply: We added the exact suggested wording in revised Introduction (Revised pg 2. Line 68, highlighted in red).

Reviewer #2 (Remarks to the Author):

Reply: Thank you for your service.

Reviewer #3 (Remarks to the Author):

General comment:

Single dots graphs should be used in all figures

Reply: Single dots graphs were used in all revised figures.

Point by point to address:

R2 to Q2: As suggested, we added unbiased quantification (see Methods) for all immunostaining shown (revised Figs. 1d, 2c, 3e, 3f, 5b, 5f, S6a, and S6a). We also quantified the colocalization of LC3 and Lamp1 (Fig. 3H), as well as the colocalization of P- α Syn and autofluorescence (Fig. 7g) by presenting the Pearson coefficient (described in revised Results). To revise: 3H reports photographs of the colocalization, but not quantifications. Please, add the graph

Reply: We showed the quantification of Fig. 3h in words as part of text. In Results, It reads “...the KO-expressed LC3- and Lamp1-labeled puncta KO retina did not overlap well (Pearson coefficient = 0.37+0.08; Fig. 3h), indicating impaired fusion of autophagy and lysosome (i.e., autophagolysosome maturation).” We skipped showing a single bar graph for the interest of space.

R5 to Q5) - We have replaced the original immunoblots with better represented ones with their quantification to demonstrate the pattern in the increase of the oligomerization of P- α Syn in KO mouse retinal lysate (in Fig. 5e,f). We add the quantification of the retina-stained P- α Syn (revised Fig. 5a,b). We add the data to show that KO retina expressed P- α Syn is resistant to proteinase-K treatment (revised Fig 7h).

To revise: Revised 7h shows one cell, which is not sufficient to prove the resistance to pk. Again, quantification is necessary and larger panels for h including more cells should be provided together with the zoom.

Quantification of the oligomer in figure e is really challenging. There is no evidence of the presence of the oligomeric form.

Reply: As suggested, we added the quantification of P-aSyn with autofluorescence (AF) in proteinase K (PK) treated samples. The bar graphs in revised Fig. S7i showing the Pearson coefficient of PK treated vs. untreated are comparable. The PK treatment unavoidably dislodged many microglia attached to the RPE apical surfaces; showing a zoom out view of stained microglia is not space-efficient.

The signals of oligomers on immunoblots are not strong but measurable. The weak signal of high molecular weight oligomers might be caused, at least in part, by technical issues, such as protein transferring from gels to blots. Higher molecular weight proteins require a higher voltage and longer transfer time, but this condition might be too strong for retaining low molecular weight proteins on blots. The compromised transferring condition was employed in our experiments to detect both low- and high-molecular weight of P-aSyn.

Q7) -R7: As suggested, we added the scotopic visual acuity test results of 2-3-month-old mice in revised Fig. S4b. The KO (vs.) showed lower visual acuity (cycles/degree; N=13 (Ctrl) N=6 (KO)), though the difference is not statistically significant.

These results are expected because, unlike ERG which measures the electrical response of the retina to light, the plasticity of brain may adapt the retinal-mediated light transduction defect; this reflects on the visual acuity behavioral task. Mice of young age do not perform consistently in this behavior test so we did not perform the visual acuity test in 1-month-old mice.

To revise: I agree with the authors regarding the interpretation of the results. However, they should be discussed in relation to previous evidence in the literature demonstrating visual acuity deficits in animal models of wild-type alpha-synuclein overexpression in the retina of control mice (Marrocco et al., Scientific Reports, 2020). Additionally, the absence of a visual acuity defect in the model represents a limitation of the study that should be clearly emphasized in the manuscript.

Reply: The referred paper (Marrocco, Indrieri et al. 2020) showed mice that had aSyn overexpressed in inner retinas (through intraocular injection of virus) exhibited significantly lower visual acuity. Perturbing inner retinal neurons, which governs retinal circuitry, possibly causes a stronger manifestation in visual acuity tests. Also, Marrocco et al., measured visual acuity using water maze task, which measures perceptual visual acuity, whereas we used optomotor system that measures reflexive optomotor responses (Jeffrey 2011).

As suggested, we added relevant comments matters in revised Discussion (highlighted in red).

Reviewer #4 (Remarks to the Author):

The authors didn't fully address my early concerns and didn't provide new evidence but new arguments. For instance, the authors reasoned that 661W cell line expresses rod-precursor, there is no other characterized photoreceptor cell line, they thus believe that 661W cell is the only choice available. However, there are three strong arguments against them. One is that what they found on the cell line is irrelevant to the function of VPS35-deficient rods in the mice. Unless, the authors can demonstrate that VPS35-deficient-661W cell and VPS35-deficient rods are the same cells. Secondly, primary rods from VPS35-deficient mice are obviously the best

choice for in vitro experiments in this study. However, the authors did not do it. Thirdly, the authors found that rod-specific VPS35 deletion causes dysregulated OS genesis, endolysosomal homeostasis, protein degradation and autophagic flux in rods. As one Reviewer mentioned (and I totally agree) that previous studies have demonstrated that VPS35 is strongly associated with autophagy, the dysregulation of autophagy, cellular toxicity, and accumulation of a-syn aggregates are expected but not unexpected. Therefore, we wonder what are new significant findings in this study.

Reply: We addressed the physiological relevance of 661W cell studies by adding the following new data.

Like in 661W cells, HSC70 and VPS35 were also partially colocalized in rhodopsin-labeled rods isolated from Ctrl mouse retinas (Fig S5d). In KO rods, HSC70 became concentrated in CD63-labeled LEs that also contain aSyn signal (Fig. S5e). Furthermore, retinal staining showed in contrast to the diffused HSC70 staining in Ctrl rods, HSC70 was prominently appeared in CD63-labeled LEs in VPS35-KO rods (Fig. S5f).

Regarding the novelty, the present work provides several advancements in the following research areas:

(1) We generated a new mouse model $Rod^{\Delta Vps35}$ manifesting Lewy body-like aSyn lesions, resembling those observed in human PD brains, in the retina. *Unlike previous PD models, the retinal aSyn-lesions of $Rod^{\Delta Vps35}$ mice are derived from the endogenous aSyn.*

The pathological manifestations in $Rod^{\Delta Vps35}$ mice are fully-penetrant, and can be longitudinally tracked using non-invasively tests like ERG, SD-OCT, and cSLO. We thus propose that this novel mouse line *has a potential for testing strategies for ameliorating the aSyn lesions and their associated neuronal degeneration.*

(2) *Our data showed the cSLO detected bright AF dots in the fundus of live $Rod^{\Delta Vps35}$ mice. These AF dots correspond to the pathological retinal microglia that had concentrated P-aSyn aggregates. The emergence and the number of the AF dots correlate to the onset and progression of retinal degeneration, respectively. These finding led us to propose that funduscope-detected AF dots are a surrogate of the aSyn pathology in synucleinopathy-associated diseases.*

(3) The dysregulation of selective microautophagy, endosomal trafficking, and protein quality control has all been linked to a-Syn aggregation. The interrelationship between the first two has long been studied, whereas little is known about the crosstalk between the latter two. Here, *we discovered the physical and functional interaction between the endolysosomal trafficking hub VPS35 and the molecular chaperon HSC70. Mechanistically, we revealed VPS35 deficiency caused late endosome sequestering of the HSC70 with aSyn. Our results show the convergence of three major pathways (i.e., selective microautophagy, endosomal trafficking, and protein quality control) that modulate the predisposition of a-Syn aggregation.*

(4) VPS35 has strong genetic and pathological links to PD and Alzheimer's of which disease feature early synapsis loss. The presynaptic function of VPS35 in mammalian neurons has not been previously reported. *This report presents morphological and functional evidence supporting the role played by VPS35 in synaptic vesicle recycling. $Rod^{\Delta Vps35}$ mice lost rod terminals preceding cell death.*

Reviewer #5 (Remarks to the Author):

The authors addressed this reviewer's requests fairly well, except for the first critique. I still believe that neuropathology analysis of the current model is required to claim this model as a synucleinopathy model. A model for synucleinopathy has to mimic the neuropathology and symptoms of the disease. Although many PD patients have visual symptoms, it is not yet fully understood whether these originated from retinal dysfunction or brain degeneration. Therefore, the authors should provide suitable neuropathology analysis to use the terms specific to a synucleinopathy model. If the authors can't analyze or determine them, the authors should use a different set of terminology to describe this model.

Reply: As suggested, we have modified the title of the manuscript by replacing "synucleinopathy" as "a-synuclein pathology". Also, in revised Discussion, we replaced the "synucleinopathy-associated pathology" to "aSyn lesions" (Line 526). We deleted synucleinopathy in the original sentence reads "The application of this novel ocular synucleinopathy model in investigating brain-related clinical histopathology and symptoms is a future interest" (Line 541). The new sentences reads "The application of this novel ocular model in investigating brain-related PD-associated clinical histopathology and symptoms is a future interest." Changes are highlighted in red.

References

- Hoang, T., J. Wang, P. Boyd, F. Wang, C. Santiago, L. Jiang, S. Yoo, M. Lahne, L. J. Todd, M. Jia, C. Saez, C. Keuthan, I. Palazzo, N. Squires, W. A. Campbell, F. Rajaii, T. Parayil, V. Trinh, D. W. Kim, G. Wang, L. J. Campbell, J. Ash, A. J. Fischer, D. R. Hyde, J. Qian and S. Blackshaw (2020). "Gene regulatory networks controlling vertebrate retinal regeneration." Science **370**(6519).
- Jeffrey, B. G., McGill, T.J., Haley, T.L., Morgans, C.W., Duvoisin, R.M. (2011). " Anatomical, Physiological, and Behavioral Analysis of Rodent Vision." In: Raber, J. (eds) Animal Models of Behavioral Analysis. Neuromethods, 50.
- Marrocco, E., A. Indrieri, F. Esposito, V. Tarallo, A. Carboncino, F. G. Alvino, S. De Falco, B. Franco, M. De Risi and E. De Leonibus (2020). "alpha-synuclein overexpression in the retina leads to vision impairment and degeneration of dopaminergic amacrine cells." Sci Rep **10**(1): 9619.

REVIEWERS' COMMENTS

Reviewer #1 (Remarks to the Author):

Authors adequately responded to all the comments and significantly improved the quality and rigor of this manuscript. A few final minor comments: 1. Figure S1c – add the merged images of Vps35 and cone arrestin to better show that the remaining Vps35 in the IS is expressed in cones; 2. Figure 1d – add the quantification of Vps35 in ONL; 3. All bar graphs –increase individual dot sizes substantially to be visible.

Reviewer #2 (Remarks to the Author):

Reviewer #3 (Remarks to the Author):

The authors answered most of my concerns. I suggest re-evaluating the optical density of the presented gel because the immunoblots presented in Figure 5e do not show clear evidence of high molecular weight oligomers. Furthermore, only 2 of the 3 KO samples appear to show an increase in monomeric forms, if any. These graphs are missing single point values. I suggest the author increase the number of samples and provide clear evidence of increased oligomeric forms.

Reviewer #5 (Remarks to the Author):

I appreciate the effort the authors made to improve their manuscript. I am satisfied with the revised manuscript.

REVIEWERS' COMMENTS

Reviewer #1 (Remarks to the Author):

Authors adequately responded to all the comments and significantly improved the quality and rigor of this manuscript. A few final minor comments: 1. Figure S1c – add the merged images of Vps35 and cone arrestin to better show that the remaining Vps35 in the IS is expressed in cones; 2. Figure 1d – add the quantification of Vps35 in ONL; 3. All bar graphs –increase individual dot sizes substantially to be visible.

Reply. As requested, (1) we added merged images of VPS35 and cone arrestin in revised Fig. S1C, (2) we added the quantification of VPS35 in ONL in revised Fig. 1d, and (3) we enlarge the individual dots in all revised figures.

Reviewer #2 (Remarks to the Author):

Reply. Thank you.

Reviewer #3 (Remarks to the Author):

The authors answered most of my concerns. I suggest re-evaluating the optical density of the presented gel because the immunoblots presented in Figure 5e do not show clear evidence of high molecular weight oligomers. Furthermore, only 2 of the 3 KO samples appear to show an increase in monomeric forms, if any. These graphs are missing single point values. I suggest the author increase the number of samples and provide clear evidence of increased oligomeric forms.

Reply: As suggested, we replaced Fig. 5e with a properly adjusted image to better show the high molecular weight oligomers. We also increased the number of samples; the revised Fig 5f (with single point values) still showed the increased both intermediate oligomers and very high molecular weight oligomers in KO albeit the heterogeneity between samples.

Reviewer #5 (Remarks to the Author):

I appreciate the effort the authors made to improve their manuscript. I am satisfied with the revised manuscript.

Reply. Thank you.